# Source attribution of black carbon and its direct radiative forcing
# in China
Yang Yang[1], Hailong Wang[1*], Steven J. Smith[2], Po-Lun Ma[1], Philip J. Rasch[1]
[1]Atmospheric Science and Global Change Division, Pacific Northwest National
Laboratory, Richland, Washington, USA
[2]Joint Global Change Research Institute, Pacific Northwest National Laboratory,
College Park, Maryland, USA
*Correspondence to yang.yang@pnnl.gov and hailong.wang@pnnl.gov

**Abstract**

The source attributions for mass concentration, haze formation, transport, and direct radiative forcing of black carbon (BC) in various regions of China are quantified in this study using the Community Earth System Model (CESM) with a source-tagging technique. Anthropogenic emissions are from the Community Emissions Data System that is newly developed for the Coupled Model Intercomparison Project Phase 6 (CMIP6). Over North China where the air quality is often poor, about 90% of near-surface BC concentration is contributed by local emissions. 35% of BC concentration over South China in winter can be attributed to emissions from North China and 19% comes from sources outside China in spring. For other regions in China, BC is largely contributed from non-local sources. We further investigated potential factors that contribute to the poor air quality in China. During polluted days, a net inflow of BC transported from non-local source regions associated with anomalous winds plays an important role in increasing local BC concentrations. BC-containing particles emitted from East Asia can also be transported across the Pacific. Our model results show that emissions from inside and outside China are equally important for the BC outflow from East Asia, while emissions from China account for 8% of BC concentration and 29% in column burden in western United States in spring. Radiative forcing estimated shows that 65% of the annual mean BC direct radiative forcing (2.2 W m$^{-2}$) in China results from local emissions, and the remaining 35% are contributed by emissions outside of China. Efficiency analysis shows that reduction in BC emissions over eastern China could benefit more on the regional air quality in China, especially in winter haze season.

## 1. Introduction

Black carbon (BC), as a component of atmospheric fine particulate matter ($PM_{2.5}$), is harmful to human health (Anenberg et al., 2011; Janssen et al., 2012). In addition to its impact on air quality, as the most efficient light-absorbing anthropogenic aerosols, BC is thought to exert a substantial influence on climate (Bond et al., 2013; IPCC, 2013; Liao et al., 2015). It can heat the atmosphere through absorbing solar radiation (Ramanathan and Carmichael, 2008), influence cloud microphysical and dynamical processes (Jacobson, 2006; McFarquhar and Wang, 2006), and reduce surface albedo through deposition on snow and ice (Flanner et al., 2007; Qian et al., 2015).

Due to accelerated urbanization and rapid economic growth, emissions of BC in China increased dramatically during recent decades. It contributed to about one fourth of the global emissions of BC in recent decades (Bond et al., 2007). Strong emissions lead to high concentrations of BC over China. Zhang et al. (2008) collected aerosol samples at eighteen stations spread over China during 2006 and reported BC concentrations in a range of 9–14 $\mu g\ m^{-3}$ at urban sites, 2–5 $\mu g\ m^{-3}$ at rural sites, and about 0.35 $\mu g\ m^{-3}$ at remote background sites. BC also exerts significant positive direct radiative forcing (DRF) at the top of the atmosphere (TOA) in China. Using the Regional Climate Chemistry Modeling System (RegCCMs), Zhuang et al. (2013) reported an annual mean BC DRF of 2–5 $W\ m^{-2}$ at TOA over eastern China and about 6 $W\ m^{-2}$ over Sichuan Basin in year 2006. Li et al. (2016) also showed a strong DRF of BC over the North China Plain and Sichuan Basin in most seasons except for spring when the strongest BC DRF with values of 4–6 $W\ m^{-2}$ shifted to southern China.

BC is the product of incomplete combustion of fossil fuels, biofuels, and open burning, such as forest and grassland fires and agricultural waste burning on fields. In the atmosphere the average lifetime of BC is only a few days, due to both wet removal and dry deposition, which is much shorter than that of long-lived greenhouse gases. In addition, BC lifetime is region dependent. BC in East Asia has a shorter lifetime than the global mean value due to a faster regional removal (H. Wang et al.,

2014), probably associated with strong precipitation during monsoon season. BC
emission reductions may benefit both mitigation of global climate change and
regional air quality (Shindell et al., 2012; Bond et al., 2013; Smith and Mizrahi, 2013),
especially in East Asia where fuel combustion emits substantial BC along with other
pollutant species. Many previous observational and/or modeling studies have
examined the source sector contributions of BC over China (Zhuang et al., 2014;
Y.-L. Zhang et al., 2015; Li et al., 2016). They found that residential heating and
industry sectors were the largest contributors to BC concentrations in China, while
biomass burning emissions from outside China were important to BC in western
China. An effective BC reduction in a receptor region would require knowing not only
the source sector that contributes the most to BC levels, but also the source
contributions from various locations within and outside the region. However, very few
previous studies have focused on the source attribution of BC concentrations in
various regions of China. Li et al. (2016) examined the contributions of emissions
inside and outside China to BC over China (with only two source regions) but did not
divide the source contributions from different regions inside China.

Pollution levels also show substantial daily to weekly variation. In recent years,

extreme wintertime hazy conditions occurred frequently in China and caused serious
air pollution, affecting more than half of the 1.3 billion people (Ding and Liu, 2014).
During one winter haze episode in 2013, BC concentrations increased up to about 20
and 8 μg m$^{-3}$ in Xi'an and Beijing over northern China, and 6 and 4 μg m$^{-3}$ in
Guangzhou and Shanghai over southern China, respectively (Y.-L. Zhang et al.,
2015). The transport of pollutants from upwind was reported to be one of the most
important contributors to local high aerosol concentrations during haze days (L. T.
Wang et al., 2014; Y. Yang et al., 2016). L. T. Wang et al. (2014) found that emissions
from northern Hebei and Beijing-Tianjin were the major contributor to particulate
matter (PM$_{2.5}$) pollution in Shijiazhuang in January 2013. Yang et al. (2016) confirmed
a connection between wind fields and PM$_{2.5}$ concentrations during winter hazy days
through model simulations and statistical analysis. They also found that weakened
winds contributed to increases in winter aerosol concentrations and hazy days over
eastern China during recent decades. As a chemically inert species, atmospheric BC
is a good tracer to investigate the source region contributions from local and non-local
emissions during polluted conditions that are related to long-range transport.

BC particles originating from East Asia can also be transported across the North

Pacific, reaching North America (Hadley et al., 2007; Ma et al., 2013a; Matsui et al.,
2013; H. Wang et al., 2014; Yang et al., 2015). Matsui et al. (2013) simulated outflow
of BC from East Asia using the Community Multiscale Air Quality (CMAQ) model and
found that anthropogenic emissions from China, biomass burning emissions from
Southeast Asia, and biomass burning emissions from Siberia and Kazakhstan
contributed 61%, 17%, and 6%, respectively, to the eastward BC flux at 150°E
averaged over 2008–2010. Hadley et al. (2007) estimated the trans-Pacific transport
of BC during April of 2004 using the Chemical Weather Forecast System (CFORS)
model and reported that, across 130°W, 75% of BC transported into North America
originated from Asia. Huang et al. (2012) simulated BC using the Sulfur Transport
and Deposition Model (STEM), and found emissions outside North America
contributed to 30–80% of column BC over North America in summer 2008. H. Wang
et al. (2014) examined the long-term (1995-2005) average global source-receptor
relationship of BC and found that BC emitted from the entire East Asia only contribute
less than 5% to the total BC burden in North America, although the contribution is up
to 40% near the west coast region. Few studies have examined the outflow from East
Asia and inflow into North America contributed from source regions in and outside
China. In addition, the emissions of BC from China increased dramatically during the
last few years, with the annual total anthropogenic emissions estimated to have
almost doubled in year 2014 compared to year 2000, shown in the newly developed
Community Emissions Data System (CEDS; Hoesly et al. 2017). Therefore, the
long-range transport of BC and source-receptor relationships could be quite different
from previous studies.

Due to its warming effect in the climate system, BC is potentially important for

climate mitigation and has drawn much attention recently. Source attribution of the
direct radiative effect of BC is likely to be different from that of near-surface
concentration and column burden due to the dependence of radiative forcing on the
vertical distribution of BC and its mixing state with other species that are influenced
by different regional sources. In this study, we use the Community Earth System
Model (CESM) with improved representations of aerosol transport and wet removal (H.
Wang et al., 2013) and a BC source-tagging technique (H. Wang et al., 2014).
Anthropogenic emissions from the newly developed CEDS inventory (Hoesly et al.,
2017), as released for the Coupled Model Intercomparison Project Phase 6 (CMIP6),
are used to examine the source attributions for mass concentration, long-range
transport, and direct radiative forcing of BC in various regions of China. We aim to
quantify: (1) source region contributions to concentrations of BC over various receptor
regions in China; (2) contributions to changes in BC concentrations under polluted
conditions; (3) source contributions to trans-boundary and trans-Pacific transport of
BC; and (4) source contributions to direct radiative forcing of BC in China.

The CESM model, emissions, and numerical experiment are described in

Section 2. Section 3 provides evaluation of the simulated concentration and aerosol
absorption optical depth of BC in China. Section 4 investigates source contributions
to near-surface concentrations, long-range transport and direct radiative forcing of BC
over various receptor regions using the BC source-tagging technique in CESM.
Section 5 summarizes these results.

**2. Methods**

We simulate the evolution and direct radiative forcing (DRF) of BC using CESM

version 1.2 (Hurrell et al., 2013). The atmospheric model in CESM is version 5 of the
Community Atmosphere Model (CAM5), with horizontal grid spacing of 1.9° latitude
by 2.5° longitude and 30 vertical layers ranging from the surface to 3.6 hPa used in
this study. The model treats the properties and processes of major aerosol species
(sea salt, mineral dust, sulfate, black carbon, primary organic matter and secondary
organic aerosol) using a three-mode modal aerosol module (MAM3), in which aerosol
size distributions are represented by three lognormal modes: Aitken, accumulation,
and coarse modes. BC is emitted to the accumulation mode. Mass mixing ratios of
different aerosol species and the number mixing ratio are predicted for each mode. A
more detailed description of the MAM3 representation can be found in Liu et al.
(2012). Aerosol dry deposition velocities are calculated using the Zhang et al. (2001)
parameterization. The wet deposition of aerosols in our CAM5 model includes
in-cloud wet removal (i.e., activation of interstitial aerosols to cloud-borne particles
followed by precipitation scavenging) and below-cloud wet removal (i.e., capture of
interstitial aerosol particles by falling precipitation particles) for both stratiform and
convective clouds. Aerosol activation is calculated with the parameterization of
Abdul-Razzak and Ghan (2000) for stratiform cloud throughout the column and
convective cloud at cloud base, while the secondary activation above convective
cloud base has a simpler treatment with an assumed maximum supersaturation in
convective updrafts (H. Wang et al., 2013). The unified treatment for convective
transport and aerosol wet removal along with the explicit aerosol activation above
convective cloud base was developed by H. Wang et al. (2013) and included in the
CAM5 version being used in this study. This implementation reduces the excessive
BC aloft and better simulates observed BC concentrations in the mid- to
upper-troposphere. Aerosol optical properties for each mode are parameterized
according to Ghan and Zaveri (2007). Refractive indices for aerosols are taken from
the OPAC (optical properties for aerosols and clouds) software package (Koepke and
Schult, 1998), but for BC at solar wavelengths the values are updated from Bond and
Bergstrom (2006). In MAM3, the aging process of BC is neglected by assuming the
immediate mixing of BC with other aerosol species. Direct radiative forcing of BC is
calculated as the difference in the top-of-the-atmosphere net radiative fluxes with and
without BC for the all-sky condition following Ghan (2013).
Anthropogenic emissions used in this study are from the CEDS dataset, as
released for the CMIP6 model experiments (Hoesly et al. 2017). This newly released
emission inventory includes aerosol (black carbon, organic carbon) and aerosol
precursor and reactive compounds (sulfur dioxide, nitrogen oxides, ammonia, carbon
monoxide, and non-methane volatile organic compounds). The emissions are
provided at monthly resolution for each year of 1750–2014 on a 0.5° x 0.5° grid and
include agricultural, energy, industry, residential, international shipping, solvents,
surface transportation, waste treatment, and aircraft sectors. The biomass burning
emissions used in this study are also developed for CMIP6 based on Global Fire
Emission Database (GFED) version 4, Fire Model Intercomparison Project (FireMIP),
visibility-observations and Global Charcoal Database (GCD) data (van Marle et al.

2016).

Figure 1a shows the horizontal spatial distribution of annual emissions of BC
averaged over the most recent 5 years (2010–2014) and the seven geographical
source regions tagged in continental China, including North China (NC), South China
(SC), Southwest China (SW), Central-West China (CW), Northeast China (NE),
Northwest China (NW), and Tibetan Plateau (TP). Figure 1b summarizes the total
seasonal BC emissions in each of these source regions. North China has the largest
annual emissions of BC in China, with maximum emission larger than 1.2 g C m$^{-2}$
year$^{-1}$ and a regional total emission of 1089 Gg C year$^{-1}$ (44% of total emissions from
continental China). Annual emissions of BC also have large values over South and
Southwest China, with maximum values in the range of 0.8–1.2 g C m$^{-2}$ year$^{-1}$,
followed by Central-West and Northeast China. Over the less economically developed
Northwest China and remote region Tibetan Plateau, emissions of BC are much
lower than other regions in China. The seasonal mean emissions of BC also show the
same spatial pattern as the annual means. BC had the largest emissions over North,
South, and Southwest China in all seasons, among which emissions are strongest in
December-January-February (DJF), especially over North China, resulting from
domestic heating. The total seasonal emissions of BC in continental China are 797,
586, 537, and 577 Gg C in DJF, March-April-May (MAM), June-July-August (JJA),
and September-October-November (SON), respectively, which add up to a total
annual BC emissions of 2497 Gg C averaged over years 2010–2014. The
anthropogenic emissions of BC in China in 2010–2014 are larger than those used in
the previous studies for earlier years (Table S1), partly as a result of a higher estimate
of BC emissions from coal coking production. The higher emissions likely lead to
higher concentrations and direct radiative forcing, and source contributions of BC in
China, compared to the values reported in these studies. The DJF emissions account
for 26–35% of annual total whereas emissions in JJA only account for 17–24% over
the seven source regions in continental China. Total BC emissions from neighboring
regions including rest of East Asia (REA, with China excluded), South Asia (SAS),
Southeast Asia (SEA), and Russia/Belarussia/Ukraine (RBU) are shown in Figure 1c.
These source regions outside China are consistent with source regions defined in the
second phase of Hemispheric Transport of Air Pollution (HTAP2). South Asia and
Southeast Asia have relatively high emissions. They may dominate the contribution to
concentrations and direct radiative forcing of BC in China, especially southern and
western China, from foreign sources through long-range transport.

An explicit BC source tagging capability was originally implemented in CAM5 by H.

Wang et al. (2014), through which emissions of BC from independent source regions
and/or sectors can be explicitly tracked. This method quantifies the source–receptor
relationships of BC in any receptor region within a single model simulation without
perturbing emissions from individual source regions or sectors. R. Zhang et al.
(2015a,b) used this method to quantify the source attributions of BC in western North
America, Himalayas, and Tibetan Plateau. The same BC source tagging technique is
implemented to a newer model version (CAM5.3) and applied in this study to quantify
the source attributions of concentration, transport and direct radiative forcing of BC in
various regions of China. BC emissions (anthropogenic plus biomass burning) from
seven geographical source regions, including North China, South China, Southwest
China, Central-West China, Northeast China, Northwest China, Tibetan Plateau in
China, and from rest of the world (RW) are tagged. Transport and physics tendencies
are calculated separately for each tagged BC in the same way as the original BC
simulation in CESM. We choose the seven individual regions (North China, South
China, Southwest China, Central-West China, Northeast China, Northwest China, and
Tibetan Plateau) and all seven regions combined (hereafter continental China) as
receptor regions in this study to examine the source-receptor relationships of BC.
While all emissions, including sulfur dioxides, organic carbon and BC, were used in
the model simulation, tagging was only applied to BC emissions.
The CAM5 simulation is performed at 1.9° × 2.5° horizontal grid spacing using the
specified-dynamics mode (Ma et al., 2013b), in which large-scale circulations (i.e.,
horizontal winds) are nudged to 6-hourly reanalysis data from the Modern Era
Retrospective-Analysis for Research and Applications (MERRA) reanalysis data set
(Rienecker et al., 2011) with a relaxation time scale of 6 hours (K. Zhang et al., 2014).
The use of nudged winds allows for a more accurate simulation so that the key role of
large-scale circulation patterns matches observations over the specified years. The
simulation is run from year 2009 to 2014, with both time-varying aerosol emissions
and meteorological fields. The first year is for spin-up and the last five years are used
for analysis.

**3. Model evaluation**
The simulations of aerosols, especially BC, using CAM5 have been extensively
evaluated against observations including aerosol mass and number concentrations,
vertical profiles, aerosol optical properties, aerosol deposition, and cloud-nucleating
properties in several previous studies (e.g., Liu et al., 2012, 2016; H. Wang et al.,
2013; Ma et al., 2013b; Jiao et al., 2014; Qian et al., 2014; R. Zhang et al., 2015a,b).
Here we focus on the evaluation of model performance in China using measurements
of near-surface BC concentrations, vertical profiles, aerosol index derived from
satellite, and aerosol absorption optical depth from the Aerosol Robotic Network
(AERONET).
**3.1 Mass concentrations and column burden of BC**
Figure 2 presents spatial distributions of simulated seasonal mean near-surface
concentrations and column burden of BC, both of which show a similar spatial pattern
to emissions of BC (Figure 1a) with the largest values over North China and the lowest
values over Northwest China and Tibetan Plateau. Near-surface model results are
taken to be the lowest model layer (from surface to 985 hPa in average). Among all
seasons, DJF has the highest BC levels, with values in the range of 6–12, 2–8, and 1–
8 µg m$^{-3}$ for near-surface concentrations and 6–12, 2–8, 1–12 mg m$^{-2}$ for column
burden over North, South, and Southwest China, respectively. In contrast, JJA has
the lowest BC concentrations over China due to the lower emissions and larger wet
scavenging associated with East Asian summer monsoon (Lou et al., 2016).
Averaged over continental China, near-surface BC concentrations are 2.5, 1.1, 0.8,
and 1.4 µg m$^{-3}$ in DJF, MAM, JJA, and SON, respectively, with seasonal variability of
50%. The column burden of BC shows smaller seasonal variability (40%), with
area-weighted average of 2.5, 1.4, 1.0, and 1.4 mg m$^{-2}$ in DJF, MAM, JJA, and SON,
respectively, in China. The magnitude, spatial distribution, and seasonal variations of
simulated near-surface BC concentrations over China are similar to those in Fu et al.
(2012) and X. Wang et al. (2013) using Intercontinental Chemical Transport
Experiment-Phase B (INTEX-B) emission inventory (Zhang et al., 2009) and those in
Li et al. (2016) using HTAP emission inventory (Janssens-Maenhout et al., 2015)
together with a global chemical transport model.

The simulated near-surface BC concentrations are evaluated here using

measurements at fourteen sites of the China Meteorological Administration
Atmosphere Watch Network (CAWNET) (Zhang et al., 2012). The locations of
CAWNET sites are shown in Figure S1a. The observational data include monthly BC
concentrations in years 2006–2007. Note that the simulated BC concentrations are
for years 2010–2014. Figure 3a compares the simulated seasonal mean near-surface
BC concentrations with those from CAWNET observations and Table S2 summarizes
the comparison in different regions, using modeled values from the grid cell
containing each observational site. Simulated BC concentrations at most sites are
within the range of one third to three times of observed values, except for Dunhuang
(94.68°E, 40.15°N) and Lhasa (91.13°E, 29.67°N) sites over western China, where
BC concentrations appear to be underestimated in the model (up to 20 times lower).
The possible bias is discussed in the following part. Over North China, simulated
concentrations are similar to observations in DJF, but underestimated in other
seasons. Over South China, the simulations do not have large biases compared to
the observed BC. However, simulated BC is underestimated in all seasons over
Southwest, Central-West, Northeast, Northwest China, and Tibetan Plateau.
Compared to the CAWNET data, the modeled near-surface BC concentrations have
a normalized mean bias (NMB) of –48%. Note that anthropogenic BC emissions went
up by a factor of 1.18 between 2006–2007 and 2010–2014. An emissions adjusted
comparison would result in an even larger underestimation. There are several
reasons that might cause low bias in this comparison. Liu et al. (2012) and H. Wang
et al. (2013) have previously found underestimation of BC concentrations over China
in CAM5 model and suggested the BC emissions may be significantly
underestimated. Using the global chemical transport model GEOS-Chem together
with emissions in 2006, Fu et al. (2012) found the simulated BC concentrations in
China were underestimated by 56%. With HTAP emissions at the year 2010 level, Li
et al. (2016) showed a low bias of 37% in simulated BC concentration in China.
Larger wet removal rate and shorter lifetime of aerosols along with the instantaneous
aging of BC in the MAM3 can also lead to the lower concentrations of BC (e.g., Wang
et al., 2011; Liu et al., 2012; H. Wang et al., 2013; Kristiansen et al., 2016).
Another potential cause for a bias in this comparison is spatial sampling bias.
Half of the CAWNET sites are located in urban areas, which will tend to have high
values near sources, whereas the modeled values represent averages over large grid
cells (R. Wang et al., 2014), as further discussed below.
The model captures well the spatial distribution and seasonal variation of BC
concentrations in China, having a statistically significant correlation coefficient of
+0.56 between simulated and observed seasonal BC concentrations over CAWNET
sites.
Figure S2 compares the observed and simulated vertical profiles of BC
concentrations in the East-Asian outflow region. The model successfully reproduces
the vertical profile of BC that was measured in March–April 2009 during the
A-FORCE field campaign and reported by Oshima et al. (2012).
**3.2 Aerosol absorption optical depth of BC**
To evaluate the simulated aerosol absorption optical depth (AAOD) of BC, the
AAOD data from AERONET (Holben et al., 2001) are used here. The locations of
AERONET sites in China are shown in Figure S1b. The observed AAOD are averaged
over years of 2010–2014 over seven sites and 2005–2010 over three sites with data
available. Most AERONET sites are over eastern and central China. AAOD of BC at
550nm are calculated by interpolating AAOD at 440 and 675 nm and removing AAOD
of dust from the retrieved AERONET AAOD following Bond et al. (2013). Figure 3b
compares the observed and simulated seasonal mean AAOD of BC at 550nm and
Table S3 summarizes the comparisons in different regions. The model has a low bias
in simulating AAOD of BC in China, smaller than the bias in near-surface
concentrations, with a NMB of –6%. As is the case with surface concentrations, this
bias could be due to model issues, such as BC transport or optical parameterization;
an underestimate in emissions; or spatial sampling bias. Simulated AAOD of BC are
within the range of one third to three times of observed values at most sites, with the
spatial distribution and seasonal variation broadly captured by the model. All but one
of the observations are located in the North and South China regions, and simulated
BC AAOD are, on average, similar to observations there. The AAOD from one
observation site in Central-West China is higher than the modeled value in DJF and
lower in other seasons. Note that, the observed AAOD of BC is derived from
AERONET measurements using the absorption Ångström exponent. A recent study
(Schuster et al., 2016) reported that absorption Ångström exponent is not a robust
parameter for separating out carbonaceous absorption in the AERONET database,
which could cause biases in the AAOD estimates.
Figure 4 shows the spatial distribution of simulated seasonal mean AAOD of total
aerosols and Aerosol Index (AI) derived from Ozone Monitoring Instrument (OMI)
measurements over years of 2010–2014. AI is a measure of absorbing aerosols
including BC and dust. Compared to satellite AI data, the model roughly reproduces
spatial distribution of total AAOD in China, with large values over North, South, and
Southwest China in all seasons. AI derived from Total Ozone Mapping Spectrometer
(TOMS) measurements (Figure S3) also shows similar pattern as simulated AAOD. It
should be noted that, besides BC, dust particles also largely contribute to AI and
produces large AI values over Northwest China.
To examine the potential model bias more broadly we compared the difference of
AAOD and AI between western and eastern China (Fig. 4). Averaging AI and AAOD
broadly over eastern and western China, we find that AAOD/AI is 0.055 over eastern
China and 0.027 over western China. If we assume that the simulated AAOD do not
have large biases over eastern China based on the evaluation against observations
shown above (Fig. 3b and Table S3), then this difference hints a possible
underestimation of BC column burden in the model over the western regions.
However, it is difficult to draw a firm conclusion, given the likely differential role of dust
in eastern vs western China. This differential likely also contributes to AAOD biases in
modeling dust and may also impact biases in the satellite derived AI values.

**4. Source contributions to BC concentrations, transport and direct radiative**
**forcing**
**4.1. Source contributions to seasonal mean BC concentrations**

Figure 5 shows the simulated spatial distribution of seasonal near-surface BC

concentrations originating from the seven tagged source regions in continental China
and all other sources from outside China (rest of the world, RW) and Table S4
summarizes these source-receptor relationships. It is not surprising that regional
emissions largely influence BC concentrations in the same region. For example,
emissions of BC from North China give 6.3 µg m$^{-3}$ of BC concentrations over North
China in DJF, whereas they only account for less than 1.8 µg m$^{-3}$ over other regions
in China. However, the relatively small amount of BC from upwind source regions can
also be a large contributor to receptor regions near the strong sources. BC emissions
from North China contribute large amount to concentrations over South, Southwest,
Central-West, and Northeast China. BC emissions from South and Southwest China
also produce a widespread impact on BC over other neighboring regions. The
impacts of BC emitted from the remaining China regions are relatively small both in
local and non-local regions due to weak emissions (Fig. 1b). All the sources in China
have the largest impact in DJF, resulting from the strong BC emissions in winter,
while emissions from outside China have the largest impact on BC over China in
MAM due to the seasonal high emission over Southeast Asia and the strong
springtime southwesterly winds.
Averaged over continental China, emissions of BC from North China produce
mean BC concentrations of 0.4–1.3 µg m$^{-3}$, followed by 0.2–0.5 µg m$^{-3}$ from South
China and 0.1–0.3 µg m$^{-3}$ from Southwest China emissions. For emissions over
Central-West China, Northeast China, Northwest China, and Tibetan Plateau, their
individual impact is less than 0.2 µg m$^{-3}$. In contrast, emissions from outside China
result in 0.12 µg m$^{-3}$ of BC concentrations in China in MAM and less than 0.06 µg m$^{-3}$
in JJA and son. The simulated source contributions to column burden of BC are
shown in Figure S4. They present a very similar spatial distribution and seasonal
variation to those of near-surface BC concentrations. However, the emissions from
outside China have a larger impact on the average column burden of BC over China
than on surface concentrations, with a magnitude of 0.4 mg m$^{-2}$ in MAM, which is
similar to that from sources in North China.
Figure 6 shows the spatial distribution of simulated relative contributions to
near-surface BC concentrations from sources in the seven regions in continental
China and those outside China by season. (The same plots for BC column burden are
shown in Figure S5.) For regions with higher emissions, their BC concentrations are
dominated by local emissions. In contrast, BC levels, especially column burden of BC,
over central and western China with lower emissions are strongly influenced by
non-local sources. Emissions from outside China can be the largest contributor to BC
over these regions. During DJF, MAM and SON, they contribute more than 70% to
both surface concentrations and column burden of BC in Tibetan Plateau, which is
important to the climate change due to the large climate efficacy of BC in snow (Qian
et al., 2011) and acceleration of snowmelt through elevated BC heat pump
mechanism (Lau et al., 2010). BC emissions from outside China also account for a
quite significant fraction of surface concentrations over Northwest and Southwest
China in MAM, which contribute to poor air quality over these regions.
Figure 7 summarizes source attribution for spatially averaged seasonal surface
BC concentrations for the seven receptor regions and continental China combined
(CN). Over North China, the majority of the BC concentrations are attributed to local
emissions in all seasons, with seasonal fractional contributions of 85–94%. Over

South China, the seasonal contributions from local emissions are in the range of 58–88%. Emissions from North China account for 35% of BC concentrations over South China in DJF, resulting from the wintertime northwesterly winds (Figure S6a), while emissions from outside China contribute about 10% in MAM due to the strong springtime biomass burning over southeast Asia and southwesterly winds transporting BC from southeast Asia to South China (Figure S6b). Southwest China has a similar level of local influence, with 47–81% of the BC concentration from local emissions, whereas 19% are due to emissions from outside China transported by westerly winds in MAM.

Non-local emissions from Southwest and North China together contribute 32–44% of BC concentration in Central-West China. North China emissions play an important role in BC concentrations over Northeast China, with relative contributions in a range of 21–30% in MAM, JJA and SON, while only 11% in DJF, which is associated with northwesterly winds in winter preventing northward transport of BC from North China to Northeast China. Over Northwest China and Tibetan Plateau, 18–34% and 46–78%, respectively, of BC originate from emissions outside China due to the low emissions over the less economically developed western China. For all of continental China as the receptor, the seasonal BC concentrations are largely attributed to the emissions from North and South China, with relative contributions ranging from 44–53% and 18–22%, respectively, followed by contributions from Southwest China (10–12%) and outside China (5–11%).

The source region contributions to column burden of BC in each receptor regions in China are shown in Figure S7. In general, impacts on the non-local BC column burden are larger than on surface concentrations because aerosol transport is relatively easier in free-troposphere than in the boundary layer (e.g., Yang et al., 2015). Column burdens of BC averaged over continental China mainly originate from emissions in North China, South China and outside China, with relative contributions ranging from 35–46%, 14–21% and 12–30%, respectively.

**4.2. Source contributions during polluted days**

Knowing the source attribution of BC during polluted days in China is important for
policy makers, which could provide an effective way for the mitigation of poor air
quality. Here, the polluted days are simply identified as days with daily concentrations
of BC higher than 90th percentile of probability density function in each receptor
regions. A total of different 45 days in winter in the 5-year simulation are identified as
polluted days for each region in China.
Figure 8 shows the DJF composite differences in near-surface BC concentrations
and winds at 850 hPa between polluted and normal days for each receptor region, and
Figure 9 summarizes the local and non-local source contributions to the differences.
When North China is under the polluted condition, BC concentrations are higher by
more than 70% compared to DJF average over North China, with a maximum increase
exceeding 5 µg m$^{-3}$. North China local emissions contribute 5.6 µg m$^{-3}$ to the
averaged increase in BC concentrations over North China during North China
polluted days, about 90% of the total increase. In winter, eastern China is dominated
by strong northwesterly winds (Figure S6a). The anomalous southerly winds during
polluted days (relative to DJF average) over North China prevent the high BC
concentrations from being transported to South China, leading to a reduced ventilation
and accumulated aerosols in North China.
Over South China, BC concentrations increase by up to 2–5 µg m$^{-3}$, in part due to
the transport from North China by anomalous northerly winds in the north part of
South China in South China polluted days. On average, contribution of North China
emissions to mean concentrations over South China increases by 2.0 µg m$^{-3}$ (60% of
total increase) during the South China polluted days.
During polluted days in Southwest China, the anomalous northeasterly winds in
the east part of Southwest China bring in BC from the highly polluted eastern China,
resulting in 2.1 µg m$^{-3}$ increase (74% of total increase) in the Southwest China, which
is much larger than the 0.7 µg m$^{-3}$ contribution from the Southwest China local
emissions.
The increase in BC concentrations during polluted days over Central-West China
is also largely influenced by the accumulation effect of the anomalous winds over
eastern and central China, which also transport BC from Southwest and eastern China
into the receptor region.
The polluted days in Northeast China are caused by both the accumulation of local
emissions due to the reduced prevailing northeasterly winds and anomalous transport
of BC from North China.
Emissions from outside China could contribute to increases in BC concentrations
over Northwest China and Tibetan Plateau during polluted days. However, during
wintertime regional polluted days in eastern and central China, the contributions of
emissions from outside China do not have a significant influence on the changes in
BC concentrations.
These results suggest that the transport of aerosols plays an important role in
increasing BC concentrations during regional polluted days in eastern and central
China. Reductions in local emissions could benefit mitigation of both local and
non-local haze in China. Emissions from outside China are not as important to hazy
pollution in eastern and central China, where haze episodes occur frequently in winter
due to relatively high anthropogenic aerosol emissions and abnormal meteorological
conditions (Sun et al., 2014; R. H. Zhang et al., 2014; Yang et al., 2016). Note that, in
this study, we only focus on the source-receptor relationships related to the wind
anomalies during polluted days. In addition to winds, changes in other meteorological
fields, such as precipitation, temperature, humidity, and planetary boundary layer
height, could also influence the contributions of local aerosols between polluted and
normal days. Although the BC emissions used in the simulation include a seasonal
variability that could cause some variations in simulated concentrations, the monthly
variability in DJF of BC emissions is less than 4% over China, which is negligible
compared to the differences in concentrations between polluted and normal days.

**4.3. Source contributions to trans-boundary and trans-Pacific transport**
Considering the large contributions of emissions from South and Southeast Asia
to MAM BC concentrations in the southwest China (Figure 6) and the large outflow of
aerosols from East Asia in springtime (Yu et al., 2008), it is valuable to examine the
inflow and outflow of BC in China. Figures S8a and S8b show the vertical distribution
of source contributions of emissions from outside China to BC concentrations
averaged over 75°–120°E and 25°–35°N, respectively, around the south boundary of
continental China in MAM. High concentrations of BC originating from South and
Southeast Asia are lifted to the free atmosphere in the south slope of Tibetan Plateau.
Then westerly winds transport these BC particles to Southwest China and South
China in both low- and mid-troposphere. Figures S8c and S8d present the
contributions of emissions from China to BC concentrations averaged over 120°–
135°E and 20°–50°N, respectively, around the east boundary of continental China. In
MAM, the northward meridional winds over 25°–35°N and the southward meridional
winds over 40°–50°N lead to the accumulation of BC in the lower atmosphere in
eastern China. Westerly winds then transport these BC out of China mostly under 500
hPa.

Figure 10 shows the spatial distribution of column burden and surface

concentrations of BC resulting from emissions in and outside China in MAM. Column
burden is used to represent the outflow in this study following previous studies (Chin et
al., 2007; Hadley et al., 2007). There are strong outflows across the Pacific Ocean
originating from emissions both in and outside China. Emissions from China contribute
0.20 mg m$^{-2}$ (or 55%) of MAM mean BC along 150°E averaged over 20°–60°N,
whereas emissions outside China contribute 0.16 mg m$^{-2}$ (or 45%). It suggests that
both emissions from China and outside China are important for the outflow from East
Asia. The yearly contribution from emissions from China to outflow from East Asia in
this study is 59%, similar to the contribution of 61% in Matsui et al. (2013) calculated
based on eastward BC mass flux using WRF-CMAQ model with INTEX-B missions.
Averaged over western United States (125°–105°W, 30°–50°N), emissions from
China account for 8% of near-surface BC concentrations and 29% in column burden in
MAM, indicating that emissions from China could have a significant impact on air
quality in western United States. More than half of the China contribution to BC over
western United States originates from eastern China (i.e., the tagged North and South
China).

**4.4. Source contributions to direct radiative forcing**

The high concentrations of BC in China could also have a significant impact on the climate system through atmospheric heating or direct radiative forcing. As shown in Figure 11, the annual mean direct radiative forcing (DRF) of BC at TOA is as high as 3–5 W m$^{-2}$ at some locations. Similar to the source attributions of BC concentrations (Figure 5) and burden (Figure S4), regional sources contribute the largest to DRF over the respective local regions. Among all the source regions in China, emissions from North, South, and Southwest China contribute the largest to local DRF of BC, with maximum DRF in a range of 3–5, 2–3, and 3–5 W m$^{-2}$, respectively. Other sources regions in China have relatively low contributions, with maximum values less than 2 W m$^{-2}$. Emissions outside China lead to 1–2 W m$^{-2}$ of DRF of BC over South, Southwest, Northwest China and Tibetan Plateau, and 0.2–1 W m$^{-2}$ over other parts of China, an effect that is quite widespread.

The total DRF of BC averaged over continental China simulated in this study is 2.20 W m$^{-2}$, larger than 0.64–1.55 W m$^{-2}$ in previous studies (Wu et al., 2008; Zhuang et al., 2011; Li et al., 2016), probably due to the different emissions in the time periods of study, as shown in Table S5. Emissions outside China have the largest contributions to DRF of BC in China compared to any of the individual source regions in China, with an averaged contribution of 0.78 W m$^{-2}$ (35%). This fractional contribution from emissions outside China is larger than 25% in Li et al. (2016), however we use different emissions, model and meteorology. Emissions from North China result in 0.55 W m$^{-2}$ (25%) of DRF of BC over China, followed by 0.30 W m$^{-2}$ (14%) and 0.28 W m$^{-2}$ (13%) from Southwest and South China, respectively. Emissions from Central-West, Northeast, Northwest China, and Tibetan Plateau taken together account for 0.29 W m$^{-2}$ (13%) of DRF of BC over China.

Figure 12a shows the seasonal mean DRF of BC averaged over China as a function of regional BC emissions. Because of high emissions, DRF of BC emitted from North China is the largest in all seasons, with values in a range of 0.5–0.8 W m$^{-2}$ averaged over China, followed by 0.2–0.5 W m$^{-2}$ from South and Southwest China. BC

from the other tagged regions in China contribute less than 0.2 W m$^{-2}$ in all seasons. In
general, BC DRF in each season is proportional to its emission rate.
Figure 12b presents the seasonal DRF efficiency of BC emitted from the tagged
regions and Table S6 summarizes these efficiencies. The variability of DRF efficiency
for forcing over China is determined by several factors, such as incoming solar
radiation (location of source regions), BC column burden and vertical distribution, and
transport out of the region. The China DRF efficiency is largest in western China (NW
and TP). This spatial pattern was also found by Henze et al. (2012). It can be
explained by the increase of multiple scattering effects and attenuation of the
transmitted radiation for large AOD (García et al., 2012). The Northeast China region
has a low China DRF efficiency due to transport eastward outside of China. The
remaining central and southern China regions have China DRF efficiencies that are
fairly consistent, varying by 20-30% about the average. The annual mean and
regional mean DRF efficiency in China is 0.88 W m$^{-2}$ Tg$^{-1}$, within the range of 0.41–
1.55 W m$^{-2}$ Tg$^{-1}$ from the previous studies (Table S5).
DRF efficiencies of BC from most regions have higher values in JJA and lower
values in DJF. This is primarily due to more incoming solar radiation in summer.
Insolation is the largest over Northwest China in JJA, together with less precipitation
than other regions, resulting in large DRF efficiency there. Global BC DRF efficiency,
particularly the annual average, is fairly similar for central, southern, and eastern
China regions (Fig. 12c, d). Global efficiency is still much higher for the western
regions.
BC emission reductions may impact mitigation of climate change and improve air
quality. To compare the relative importance of climate and air quality effects of BC
from different regions in China, Fig. 13 shows the near-surface concentration and
column burden efficiency of BC over China and globally and Table S7 summarizes
these efficiencies. For near-surface concentration (Fig. 13a and 13b), the efficiencies
are largest in DJF and lowest in JJA, in contrast to the DRF efficiencies, resulting
from the less precipitation and wet deposition of aerosols in winter. Unlike the DRF
efficiencies, the near-surface concentration efficiencies over eastern China are
similar and even larger than those for central and western China. These results
suggest that reduction in BC emissions in eastern China could benefit more on the
regional air quality in China, especially in winter haze season.

The relative distributions of column burden efficiencies (Fig. 13c and 13d) are

similar to the DRF efficiencies for the major emitting region in China, indicating that
aerosol lifetime in atmosphere drives DRF that influences regional and global climate.
The western regions (NW and TP), as expected, have a higher forcing per unit
column burden.

**5. Conclusions and discussions**

In this study, the Community Earth System Model (CESM) with a source-tagging

technique is used to quantify the contributions of BC emitted from seven regions in
continental China, including North China (NC), South China (SC), Southwest China
(SW), Central-West China (CW), Northeast China (NE), Northwest China (NW), and
Tibetan Plateau (TP), and sources outside China (RW) to concentrations, haze
formation, trans-boundary and trans-Pacific transport, and direct radiative forcing
(DRF) of BC in China. The anthropogenic emissions of BC for years 2010-2014 used
in this study were developed for the Coupled Model Intercomparison Project Phase 6
(CMIP6) from the Community Emissions Data System (CEDS). The annual total
emission of BC from continental China is 2497 Gg C averaged over years 2010–2014.
The model captures well the spatial distribution and seasonal variation in China.
AAOD compares well with measurements, which are largely located in central and
eastern China. Surface BC concentrations are underestimated by 48% compared to
point observations.

The individual source regions are the largest contributors to their local BC

concentration levels. Over North China where the air quality is often poor, about 90%
of near-surface BC concentration is contributed by local emissions. However, some
source regions also impact BC in neighboring regions. Due to the seasonal variability
of winds and emission rates, emissions from North China account for 35% of
near-surface BC concentrations over South China in DJF
(December-January-February), while emissions from outside China contribute about
10% in MAM (March-April-May). Over Southwest China, 19% of BC in MAM comes
from sources outside China. Southwest and North China emissions contribute largely
to BC in Central-West China. North China emissions have a contribution in a range of
21–30% to BC concentrations in Northeast China. Over Northwest China and Tibetan
Plateau, more than 20% and 40% of BC, respectively, originates from emissions
outside China. These indicate that, for regions with high emissions, their BC
concentrations are dominated by local emissions. In contrast, BC levels over central
and western China with lower emissions are more strongly influenced by non-local
emissions. For all continental China as a whole, seasonal BC concentrations are
largely due to emissions from North and South China, with relative contributions
ranging from 44–53% and 18–22%, respectively, followed by contributions from
Southwest (10–12%) and outside China (5–11%).
Emissions from non-local sources together with abnormal winds are one of the
important factors contributing to high winter time pollution events in China. Over
South China, about 60% of the increase in BC concentrations during high pollution
conditions results from North China emissions. The increases in BC concentrations
during polluted days over Southwest, Central-West and Northeast China are strongly
influenced by emissions from eastern China. Emissions from outside China could
contribute significantly to increases in BC concentrations over Northwest China and
Tibetan Plateau during their polluted days. However, emissions from outside China do
not have a significant contribution to haze in eastern and central China, suggesting
that reduction in emissions within China would be needed to mitigate both local and
non-local BC concentrations under high-polluted conditions.
Emissions from regions in and outside China both account for about half of BC
outflow from East Asia, suggesting that emissions from China and other regions are
equally important for the BC outflow from East Asia. Through long-range transport,
emissions from China result in 8% of near-surface BC concentration and 29% in
column burden over western United States in MAM, indicating that emissions from
China could have an impact on air quality in western United States.
The total DRF of BC averaged over continental China simulated in this study is
2.20 W m$^{-2}$. Among the tagged regions, emissions outside China have the largest
single contribution to DRF of BC in China, with an average contribution of 35%,
followed by 25%, 14%, and 13% due to emissions from North, South and Southwest
China, respectively. DRF efficiencies over eastern China are small compared to
central and western China in all seasons. For near-surface concentration, the
efficiencies are largest in DJF and lowest in JJA, and efficiencies over eastern China
are similar and even larger than central and western China. These suggest that
reduction in BC emissions over eastern China could benefit more on the regional air
quality in China, especially in winter haze season.
Note that the model largely underestimates BC concentrations over China,
compared to the observation, which has also been reported in many previous studies
using different models and different emission inventories (e.g., Liu et al., 2012; Fu et
al., 2012; Huang et al., 2013; H. Wang et al., 2013; Q. Wang et al., 2014; R. Wang et
al., 2014; Li et al., 2016). One possible reason is that in situ measurements are point
observations, while the model does not treat the subgrid variability of aerosols and
assumes aerosols are uniformly distributed over the grid cell. R. Wang et al. (2014)
found a reduction of negative bias (from −88% to −35%) in the modeled surface BC
concentrations when using high-resolution emissions and modeling at 0.5° X 0.7°
resolution. This result indicates that the siting of observational stations can result in
an artificial bias when comparing with relatively coarse model results. Further
investigation of this siting/resolution bias is warranted, including investigation on
whether this type of bias might extend, presumably to a lesser extent, also to AAOD
measurements.
Further reasons that could contribute to this bias are emission underestimation or
inaccurate aerosol processes in the model. Given that the differences between
modeled and observed AAOD over eastern China are relatively small (–6%), we
conclude that, given current evidence, the total amount of atmospheric BC in these
simulations is reasonable at least in this sub-region.
Over eastern China, the BC concentrations are dominated by local emissions in
this study, with local contribution of 58–94%. The underestimation of simulated BC
concentrations over eastern China is more likely due to either underestimation of
local emissions, too much aerosol removal within these regions, or resolution bias
between observations and model grids. Over western China, 18–78% of the BC
originates from emissions outside China. Thus biases of simulated BC concentrations
could also come from underestimation of emissions outside China and or too much
removal of BC during long-range transport. Satellite data are a promising method to
validate modeling and emissions inventories, given that they do not depend on the
location of observing stations, providing more uniform spatial coverage. A
comparison of modeled AAOD and satellite AI provides an indication that the
modeled burden in western China is underestimated, although the role of dust needs
to be better characterized.
Uncertainty in China BC emissions has been estimated as –43% to 93% by Lu et
al. (2011), –50% to 164% by Qin and Xie (2012), ±176% by Kurokawa et al. (2013),
and –28 to 126% by Zhao et al. (2013). The BC emissions estimates used here for
China in 2010 are 40% higher than those of Zhao et al. (2013) and Lu et al. (2011)
and 30% higher than Klimont et al. (2016), in large part due to a higher estimate of
BC emissions from coal coke production. Emissions from coke production are
particularly uncertain given that "there are no measurements for $PM_{2.5}$ and BC
emissions" (Huo et al. 2012) available to guide inventory estimates. Total rest of the
world emissions other than China, which appear to be a major contributor to burdens
over western regions, are within 1% of those from Klimont et al. (2016).
BC aging in the atmosphere is important for BC concentration and its optical
properties, which transforms BC from hydrophobic aggregates to hydrophilic particles
coated with soluble materials (Cheng et al., 2006). He et al. (2015, 2016a) found that
BC optical properties varied by a factor of two or more due to different coating
structures during BC aging process based on their theoretical and experimental
intercomparison. Oshima et al. (2009) and He et al. (2016b) pointed out that the use of
various microphysical BC aging schemes could significantly improve simulations of
BC concentrations, compared to the simplified aging parameterizations. Liu et al.
(2012) also reported that the wet removal rate of BC simulated in standard CAM5 is 60%
higher than AeroCom multi-model mean due to the rapid or instantaneous aging of BC.
H. Wang et al. (2013) showed that the explicit treatment of BC aging process with slow
aging assumptions in CAM5 could significantly increase BC lifetime and the efficiency
of BC long-range transport. In the three-mode aerosol module (MAM3) of CAM5 used
in this study, the aging process of BC is neglected by assuming the immediate internal
mixing of BC with other aerosol species in the same mode. This assumption could
lead to an overestimation of wet removal of BC and, therefore, an underestimation of
BC concentrations, absorption optical depth (Fig. 3) and direct radiative forcing. In
addition, the internally-mixed optical treatment in CAM5 could also cause bias in BC
absorption calculation. However, H. Wang et al. (2014) examined source-receptor
relationships for BC under the different BC aging assumptions and found that the
quantitative source attributions varied slightly while the qualitative source-receptor
relationships still hold. Therefore, although the magnitude of simulated BC and its
optical properties could be underestimated due to the instantaneous aging of BC and
uncertainty in coating structures, we expect that the aging treatment in MAM3 of
CAM5 should not influence the qualitative source attributions examined in this study.

In this study, BC is used as an indicator of pollution (or air quality) in China.

Although BC is often co-emitted with other species, such as primary organic matter,
organic gases and sulfuric gases, source-receptor relationship of BC may not fully
represent that of total aerosols. The contribution of BC to total near-surface $PM_{2.5}$
concentrations averaged over China is less than 10%. Other aerosols, such as sulfate,
are dominant in China during polluted days. The spatio-temporal variations and
source contributions of these species are largely different from those of BC because
spatial distributions of emissions (e.g., $SO_2$) and formation processes can be
considerably different. For example, Matsui et al. (2009) showed that primary aerosols
around Beijing were determined by emissions within 100 km around Beijing within the
preceding 24 hours, while emissions as far as 500 km and within the preceding 3 days
were found to affect secondary aerosols in Beijing. Thus, the secondary aerosols
could have larger contributions from non-local emissions than BC. BC concentrations
are highest in winter over China due to higher emissions, while sulfate concentrations
reach maximum in summer when the strong sunlight and high temperature favor the
sulfate formation. Therefore, knowing the accurate source attributions of air pollution
in China requires source tagging for more aerosol species, such as sulfate.

*Acknowledgments*.
This research was supported by the National Atmospheric and Space
Administration's Atmospheric Composition: Modeling and Analysis Program
(ACMAP), award NNH15AZ64I. We also acknowledge additional support from the
U.S. Department of Energy (DOE), Office of Science, Biological and
Environmental Research. The Pacific Northwest National Laboratory is
operated for DOE by Battelle Memorial Institute under contract
DE-AC05-76RLO1830. The CESM project was supported by the National Science
Foundation and the DOE Office of Science. The satellite-derived Total Ozone
Mapping Spectromenter Aerosol Index monthly data sets are obtained from the Web
site at http://disc.sci.gsfc.nasa.gov/data-holdings/PIP/aerosol_index.shtml. The
National Energy Research Scientific Computing Center (NERSC) provided
computational resources. Model results are available through NERSC upon request.

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

 **Figure Captions**

 **Figure 1.** (a) Spatial distribution of annual mean total emissions (anthropogenic plus

 biomass burning, units: g C m$^{-2}$ yr$^{-1}$) of black carbon (BC) averaged over 2010–2014.

 The geographical BC source regions are selected as North China (NC, 109°E–east

 boundary, 30°–41°N), South China (SC, 109°E–east boundary, south boundary–

 30°N), Southwest China (SW, 100°–109°N, south boundary–32°N), Central-West

 China (CW, 100°–109°N, 32°N–north boundary), Northeast China (NE, 109°E–east

 boundary, 41°N–north boundary), Northwest China (NW, west boundary–100°E,

 36°N–north boundary), and Tibetan Plateau (TP, west boundary–100°E, south

 boundary–36°N) in China and regions outside of China (RW, rest of the world). (b)

 Seasonal mean total emissions (units: Gg C, Gg = 10$^9$g) of BC from the seven BC

 source regions in China and emissions from rest of East Asia (REA, with China

 excluded), South Asia (SAS), Southeast Asia (SEA), and Russia/Belarussia/Ukraine

 (RBU).

 **Figure 2.** Simulated seasonal mean near-surface concentrations (left, units: μg m$^{-3}$)

 and column burden (right, units: mg m$^{-2}$) of BC in December-January-February (DJF),

 March-April-May (MAM), June-July-August (JJA), and

 September-October-November (SON).

 **Figure 3.** Comparisons of observed and modeled seasonal mean (a) near-surface

 concentrations (units: μg m$^{-3}$) and (b) aerosol absorption optical depth (AAOD) of BC

 in China. Solid lines mark the 1:1 ratios and dashed lines mark the 1:3 and 3:1 ratios.

 Observed BC concentrations were taken between 2006 and 2007 at 14 sites of the

 China Meteorological Administration (CMA) Atmosphere Watch Network (CAWNET)

 (Zhang et al., 2012). Observed AAOD of BC are obtained by removing dust AAOD

 from total AAOD at 10 sites of the Aerosol Robotic Network (AERONET) (Holben et

 al., 2001), following Bond et al. (2013). The observed AAOD are averaged over years

 of 2005–2014 with data available. Correlation coefficient (R) and normalized mean

bias (NMB) between observation and simulation are shown on top left of each panel.
$NMB = 100\% \times \sum(M_i - O_i) / \sum O_i$, where $M_i$ and $O_i$ are the modeled and observed
values at site $i$, respectively. Site locations are shown in Figure S1a.

**Figure 4.** Spatial distribution of seasonal mean AAOD of total aerosols (left) and
Aerosol Index (AI) derived from Ozone Monitoring Instrument (OMI) measurements
over years of 2010–2014 (right).

**Figure 5.** Spatial distribution of seasonal mean near-surface concentrations of BC
($\mu g\ m^{-3}$) originating from the seven source regions in China (NC, SC, SW, CW, NE,
NW, and TP), marked with black outlines, and sources outside China (RW).
Regionally averaged BC in China contributed by individual source regions is shown at
the bottom right of each panel.

**Figure 6.** Spatial distribution of relative contributions (%) to seasonal mean
near-surface BC concentrations from each of the tagged source regions.

**Figure 7.** Relative contributions (%) from the tagged source regions (denoted by
colors) to regional mean surface concentrations of BC over seven receptor regions in
China (NC, SC, SW, CW, NE, NW, and TP) and China (seven regions combined, CN)
in different seasons. The receptor regions are marked on the horizontal axis in each
panel.

**Figure 8.** Composite differences in winds at 850 hPa ($m\ s^{-1}$) and near-surface BC
concentrations ($\mu g\ m^{-3}$) between polluted and normal days in DJF.

**Figure 9.** Composite differences in surface BC concentrations ($\mu g\ m^{-3}$) averaged
over receptor regions (marked on the horizontal axis) over eastern and central China
between polluted and normal days in DJF originating from individual sources regions
(bars in each column).

**Figure 10.** Spatial distribution of (a, b) column burden (mg m$^{-2}$) and (c, d)
near-surface concentrations (µg m$^{-3}$) of BC originating from total emissions inside
(CN) and outside China (RW), respectively, in March-April-May (MAM). The black
solid lines over western (150°E, 20°–60°N) Pacific in panel (a) mark the
cross-sections used to quantify outflow of BC from East Asia. The box over western
United States (125°–105°W, 30°–50°N) in panel (c) is used to quantify BC
concentrations attributed to sources from China.

**Figure 11.** Spatial distribution of annual mean direct radiative forcing of BC (W m$^{-2}$) at
the top of the atmosphere originating from the tagged BC source regions in China
(NC, SC, SW, CW, NE, NW, and TP) and source outside China (RW). Regionally
averaged forcing in China contributed by individual source regions is shown at the
bottom right of each panel.

**Figure 12.** (a, c) BC seasonal DRF averaged over China as a function of BC
emission fraction (the ratio of regional emission to the total emission over China and
global, respectively, unit: %) for each of the tagged regions. (b, d) Seasonal DRF
efficiency of BC (W m$^{-2}$ Tg$^{-1}$) for each of the tagged source regions over China and
globally, respectively. The efficiency is defined as the DRF divided by the
corresponding scaled annual emission (seasonal emission multiplied by 4). Error bars
indicate 1-σ of mean values during years 2010–2014.

**Figure 13.** Seasonal (a, b) near-surface concentration (µg m$^{-3}$ Tg$^{-1}$) and (c, d) column
burden (mg m$^{-2}$ Tg$^{-1}$) efficiency of BC for each of the tagged source regions over
China and globally, respectively.


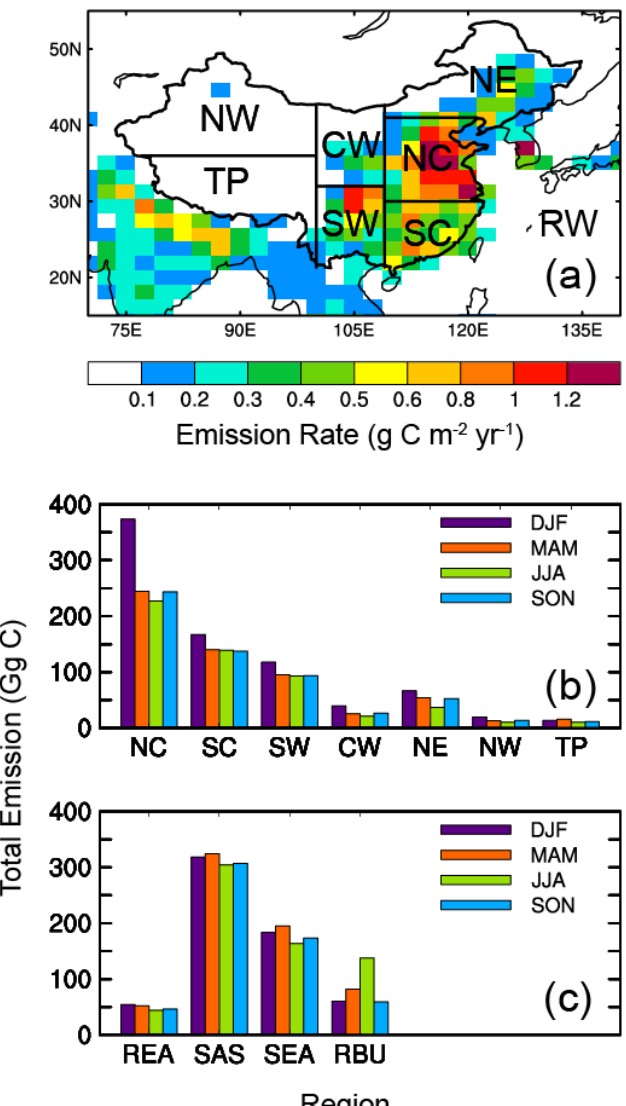

**Figure 1.** (a) Spatial distribution of annual mean total emissions (anthropogenic plus biomass burning, units: g C m$^{-2}$ yr$^{-1}$) of black carbon (BC) averaged over 2010–2014. The geographical BC source regions are selected as North China (NC, 109°E–east boundary, 30°–41°N), South China (SC, 109°E–east boundary, south boundary–30°N), Southwest China (SW, 100°–109°N, south boundary–32°N), Central-West China (CW, 100°–109°N, 32°N–north boundary), Northeast China (NE, 109°E–east boundary, 41°N–north boundary), Northwest China (NW, west boundary–100°E, 36°N–north boundary), and Tibetan Plateau (TP, west boundary–100°E, south boundary–36°N) in China and regions outside of China (RW, rest of the world). (b) Seasonal mean total emissions (units: Gg C, Gg = 10$^9$g) of BC from the seven BC

source regions in China and (c) emissions from rest of East Asia (REA, with China
excluded), South Asia (SAS), Southeast Asia (SEA), and Russia/Belarussia/Ukraine
(RBU).

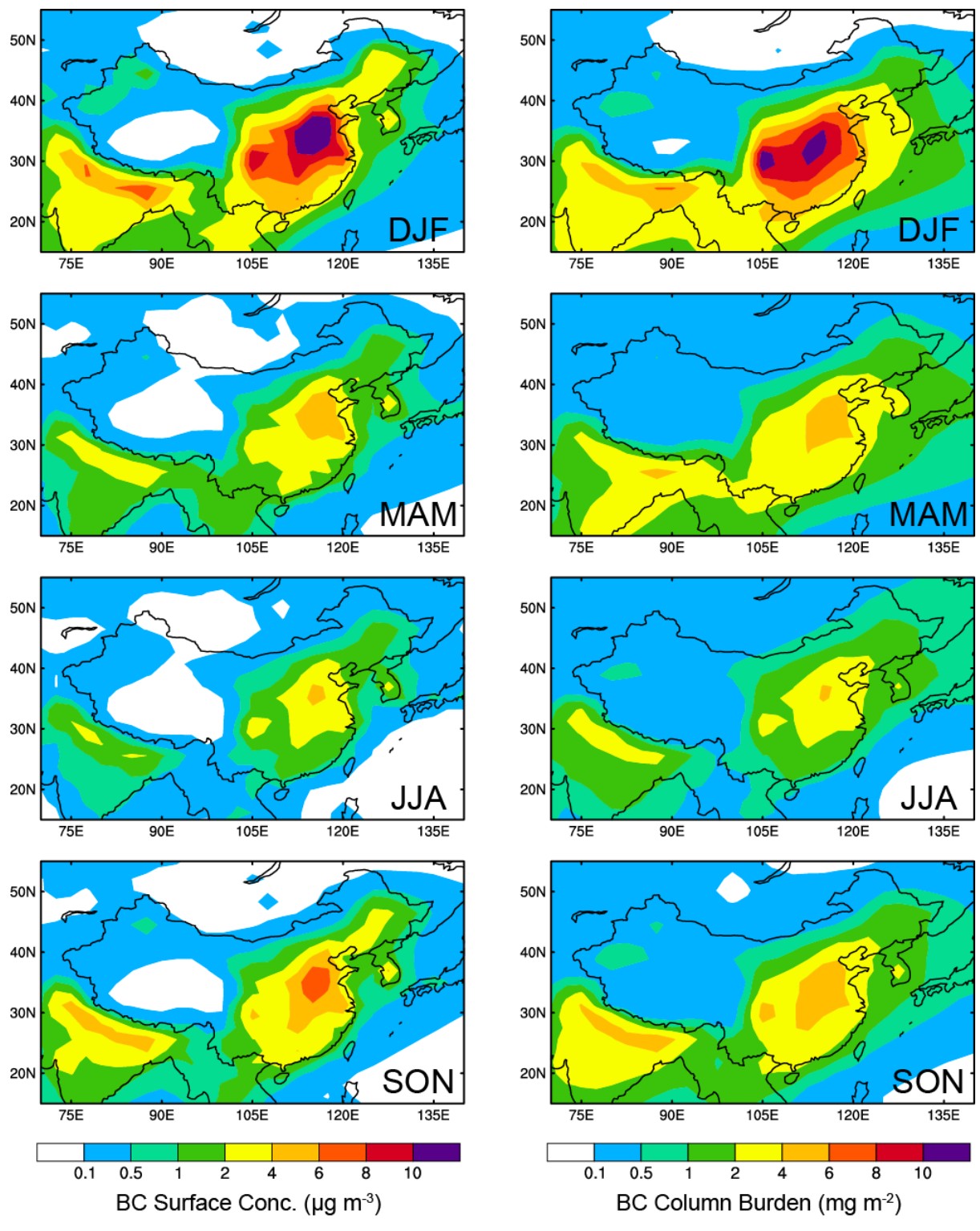



**Figure 2.** Simulated seasonal mean near-surface concentrations (left, units: μg m$^{-3}$)
and column burden (right, units: mg m$^{-2}$) of BC in December-January-February (DJF),
March-April-May (MAM), June-July-August (JJA), and
September-October-November (SON).

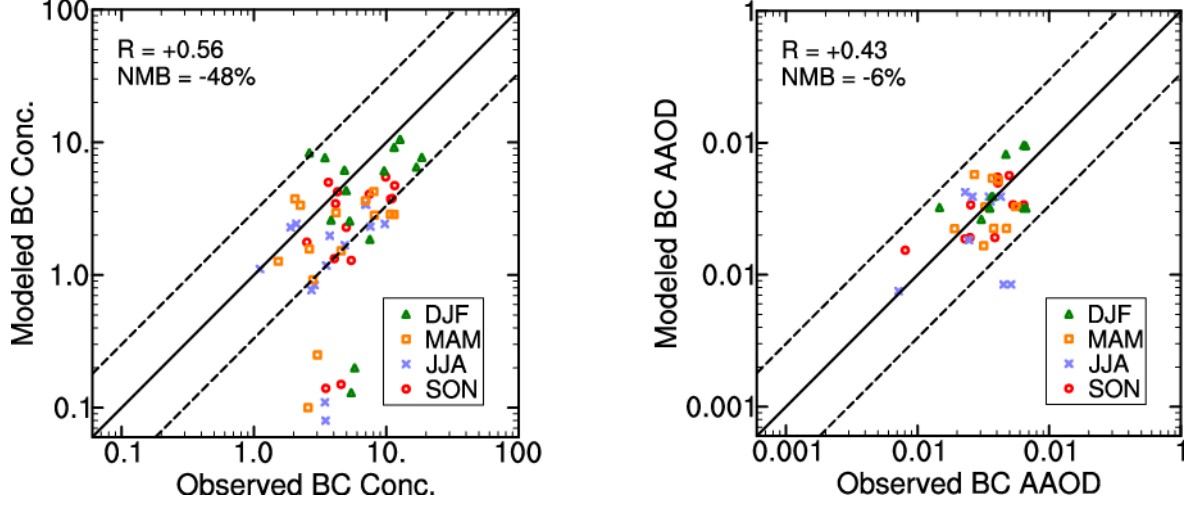



**Figure 3.** Comparisons of observed and modeled seasonal mean (a) near-surface
concentrations (units: μg m$^{-3}$) and (b) aerosol absorption optical depth (AAOD) of BC
in China. Solid lines mark the 1:1 ratios and dashed lines mark the 1:3 and 3:1 ratios.
Observed BC concentrations were taken between 2006 and 2007 at 14 sites of the
China Meteorological Administration (CMA) Atmosphere Watch Network (CAWNET)
(Zhang et al., 2012). Observed AAOD of BC are obtained by removing dust AAOD
from total AAOD at 10 sites of the Aerosol Robotic Network (AERONET) (Holben et
al., 2001), following Bond et al. (2013). The observed AAOD are averaged over years
of 2010–2014 over 7 sites and 2005–2010 over 3 sites with data available.
Correlation coefficient (R) and normalized mean bias (NMB) between observation
and simulation are shown on top left of each panel.  NMB = 100%×$\sum(M_i - O_i)/\sum O_i$,
where $M_i$ and $O_i$ are the modeled and observed values at site $i$, respectively. Site
locations are shown in Figure S1a.


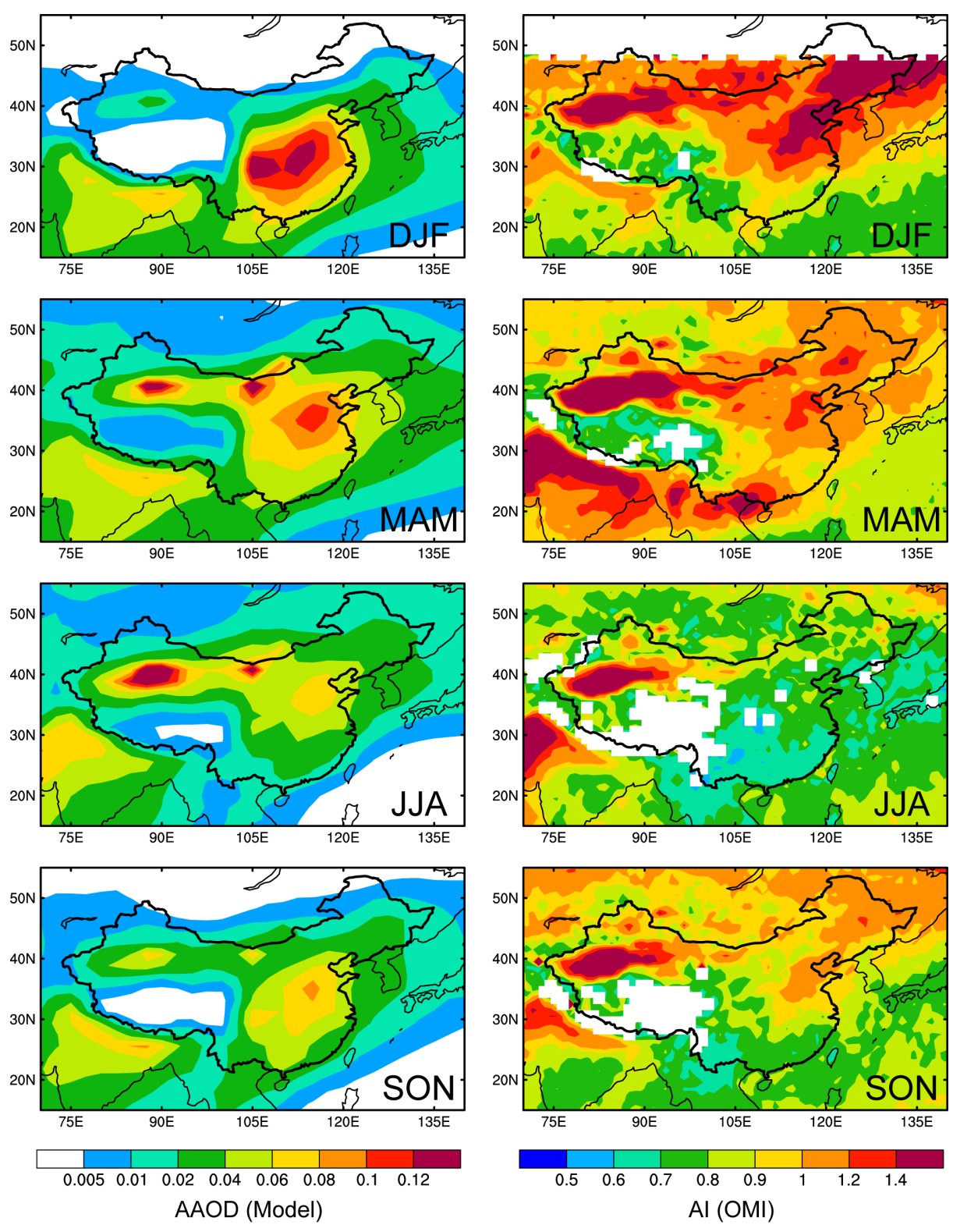



**Figure 4.** Spatial distribution of seasonal mean AAOD of total aerosols (left) and

Aerosol Index (AI) derived from Ozone Monitoring Instrument (OMI) measurements

over years of 2010–2014 (right).

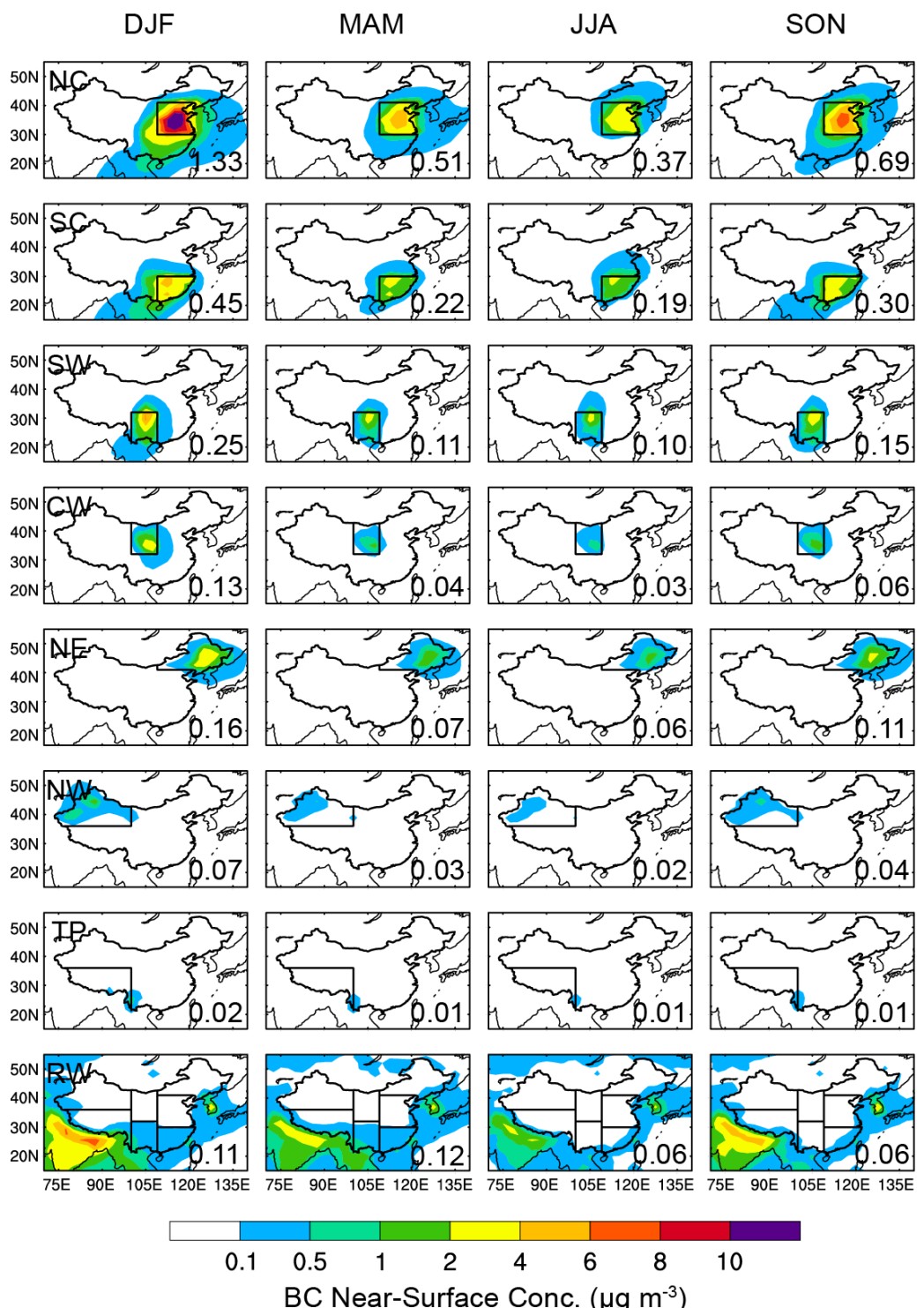

**Figure 5.** Spatial distribution of seasonal mean near-surface concentrations of BC

(μg m⁻³) originating from the seven source regions in China (NC, SC, SW, CW, NE,

NW, and TP), marked with black outlines, and sources outside China (RW).

Regionally averaged BC in China contributed by individual source regions is shown at

the bottom right of each panel.

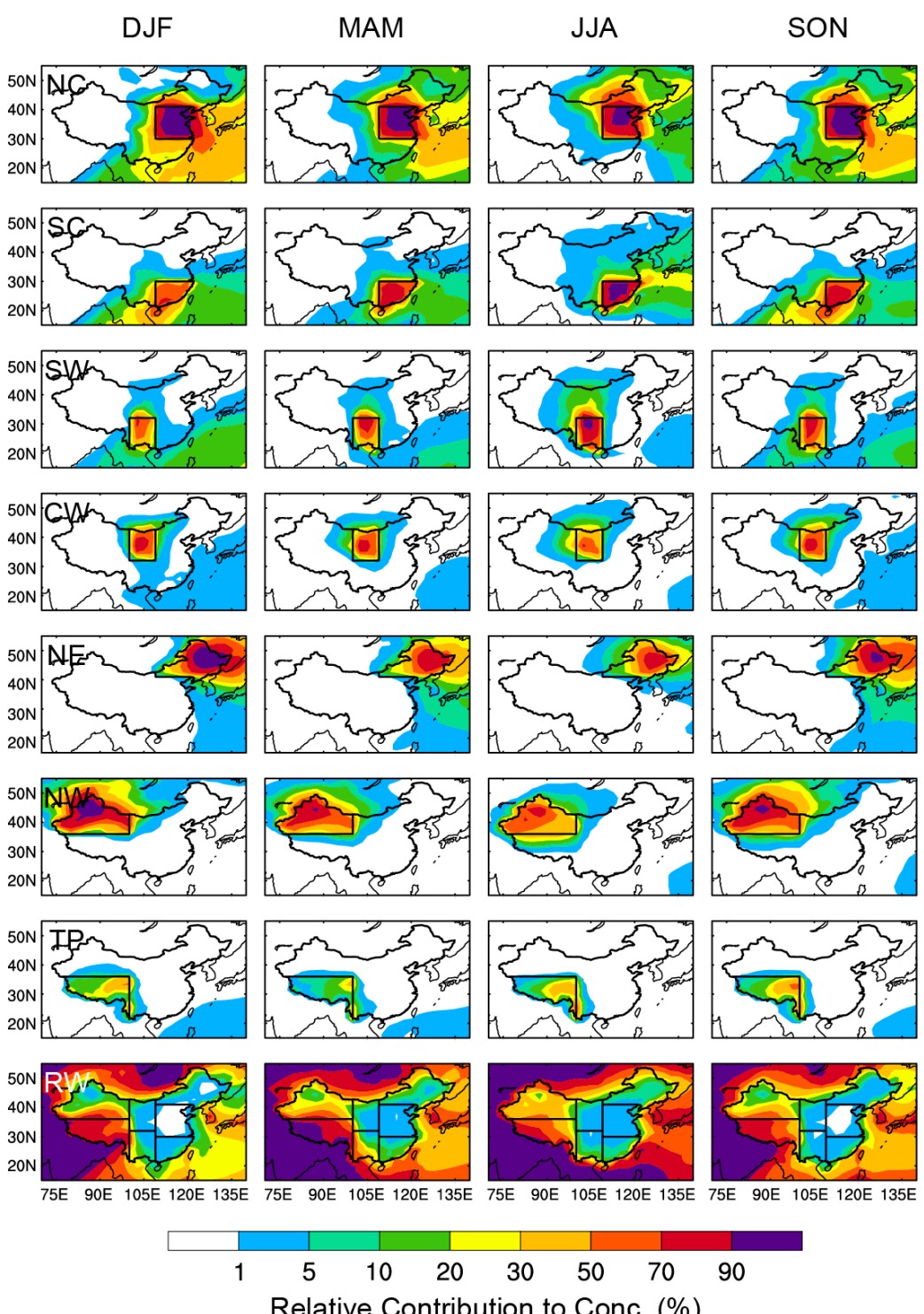

**Figure 6.** Spatial distribution of relative contributions (%) to seasonal mean near-surface BC concentrations from each of the tagged source regions.

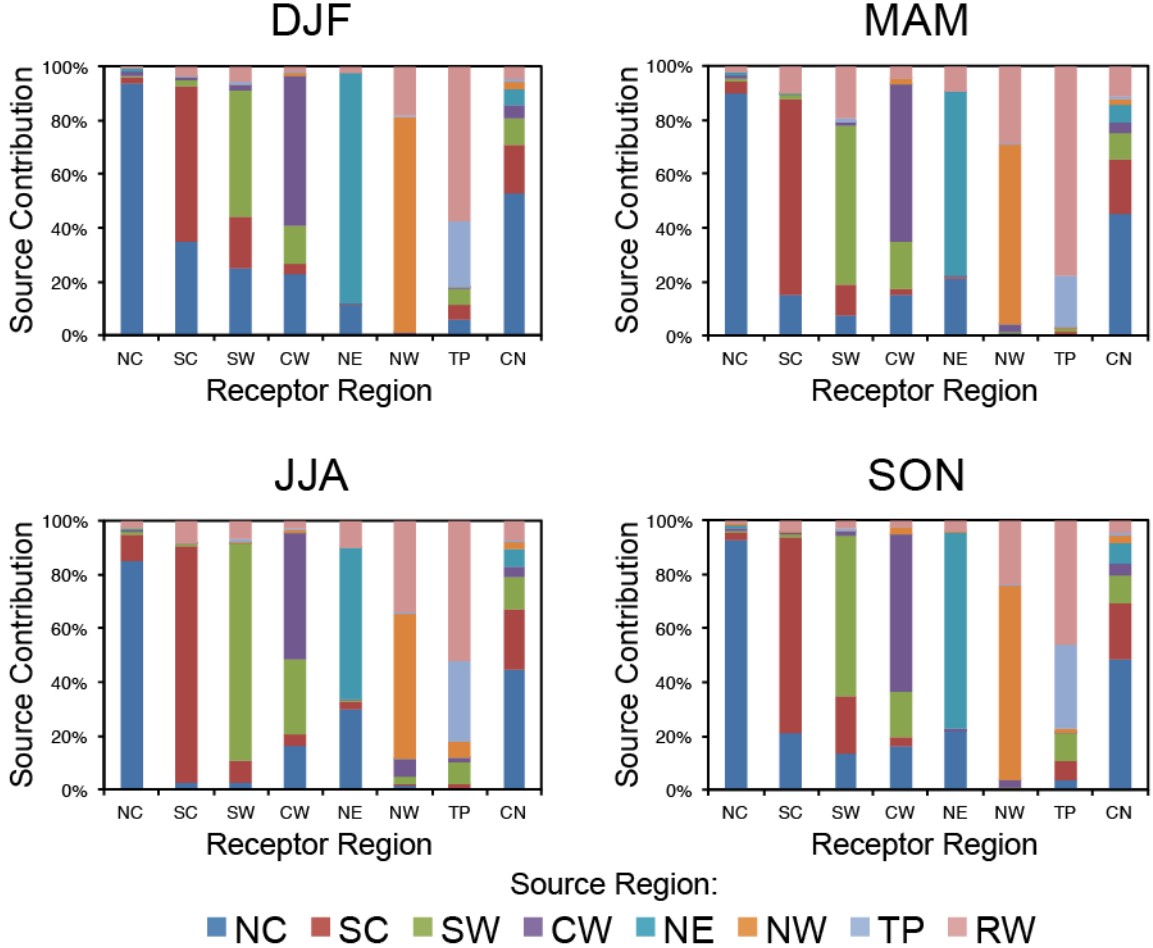



**Figure 7.** Relative contributions (%) from the tagged source regions (denoted by
colors) to regional mean surface concentrations of BC over seven receptor regions in
China (NC, SC, SW, CW, NE, NW, and TP) and China (seven regions combined, CN)
in different seasons. The receptor regions are marked on the horizontal axis in each
panel.


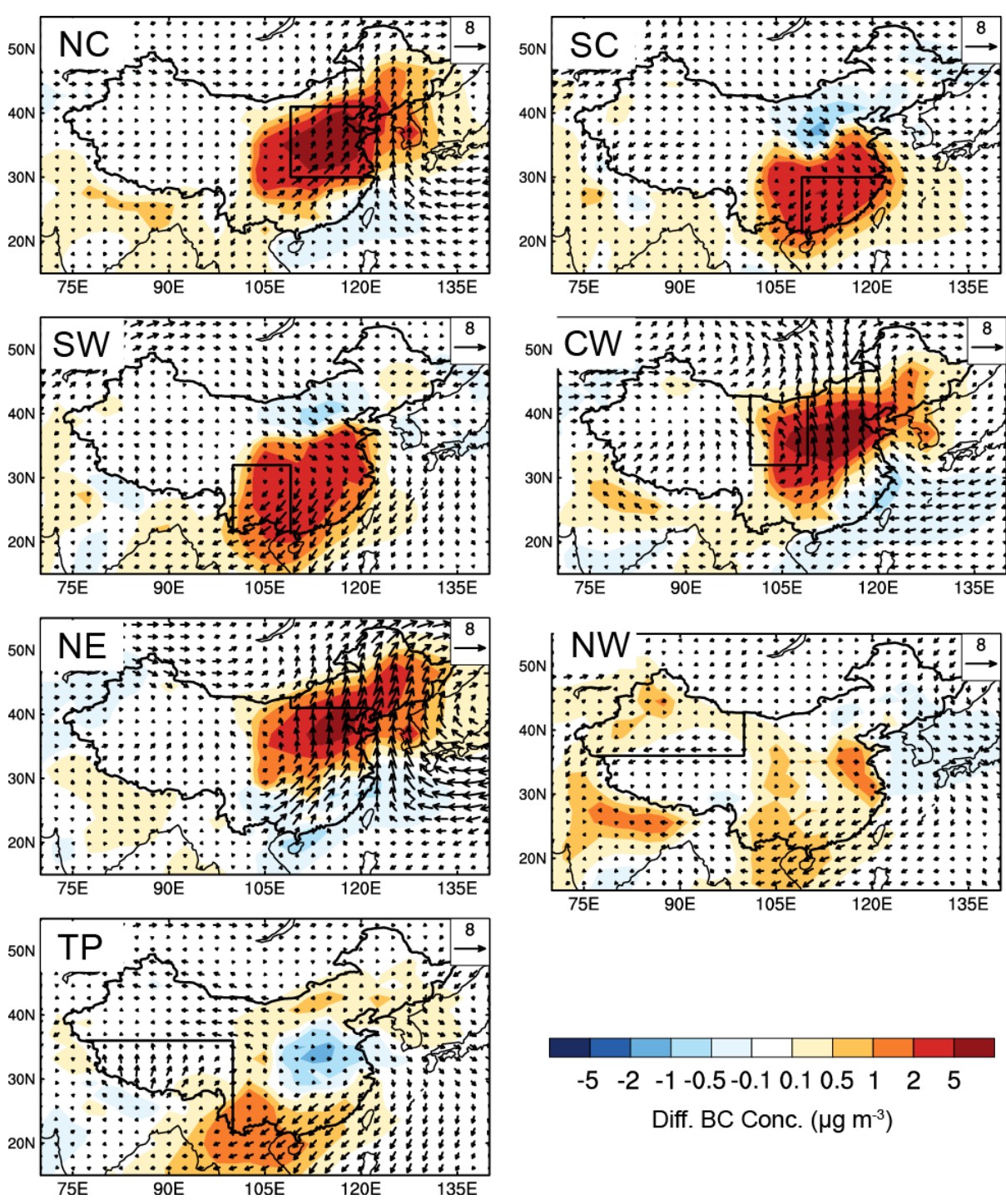



**Figure 8.** Composite differences in winds at 850 hPa (m s$^{-1}$) and near-surface BC
concentrations (μg m$^{-3}$) between polluted and normal days in DJF.

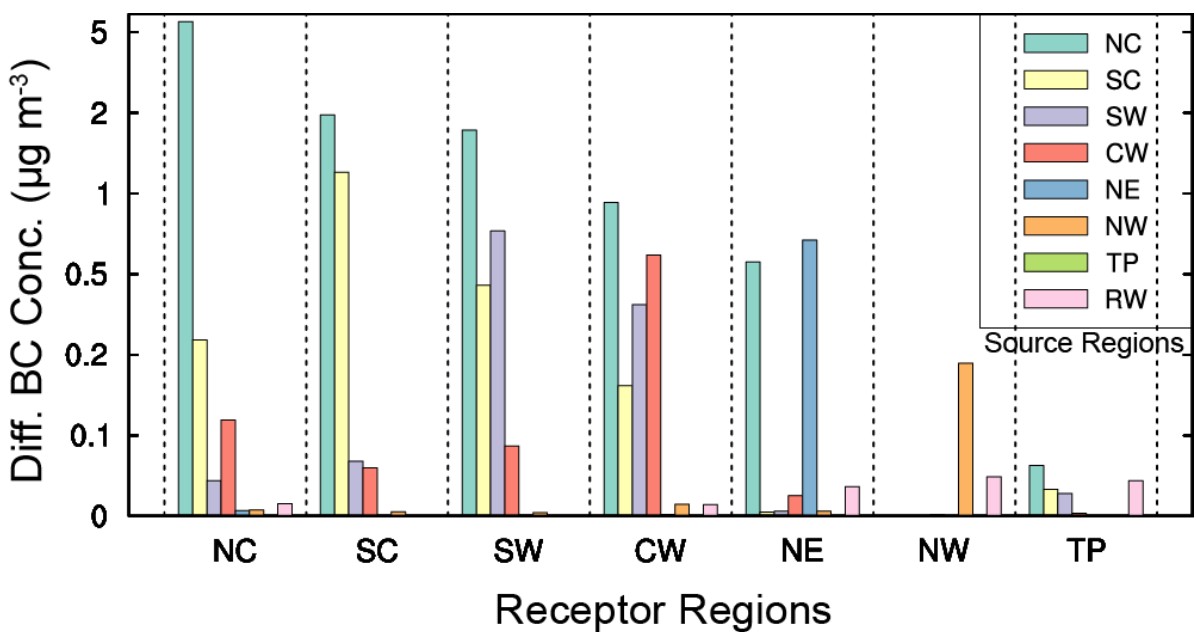

**Figure 9.** Composite differences in surface BC concentrations (µg m$^{-3}$) averaged over receptor regions (marked on the horizontal axis) over eastern and central China between polluted and normal days in DJF originating from individual sources regions (bars in each column).

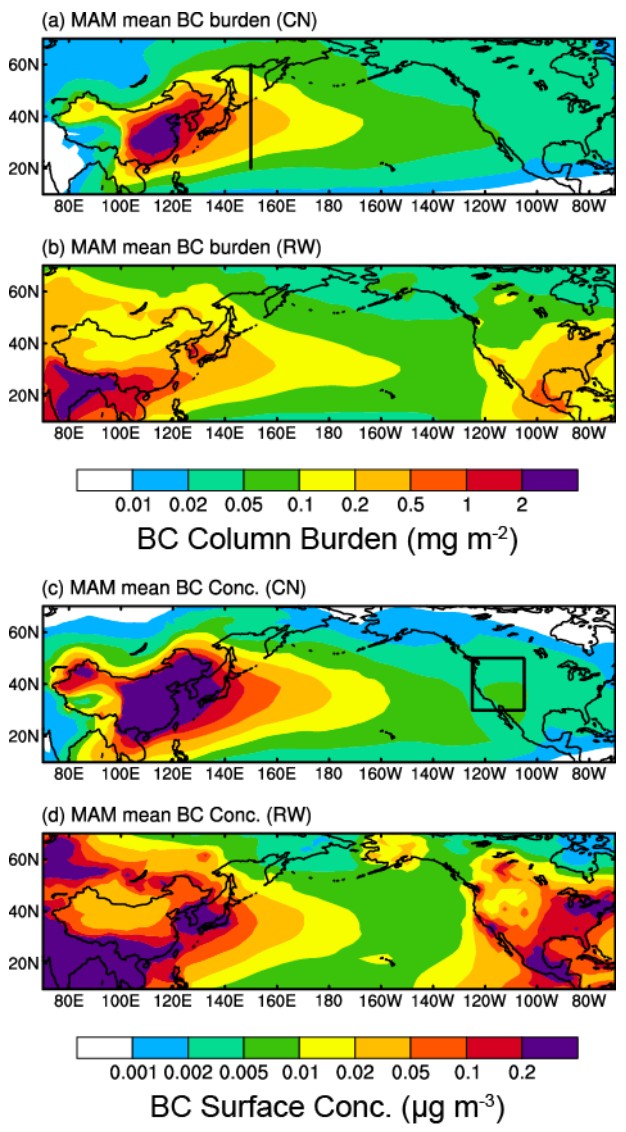

**Figure 10.** Spatial distribution of (a, b) column burden (mg m$^{-2}$) and (c, d)
near-surface concentrations (µg m$^{-3}$) of BC originating from total emissions inside
(CN) and outside China (RW), respectively, in March-April-May (MAM). The black
solid lines over western (150°E, 20°–60°N) Pacific in panel (a) mark the
cross-sections used to quantify outflow of BC from East Asia. The box over western
United States (125°–105°W, 30°–50°N) in panel (c) is used to quantify BC
concentrations attributed to sources from China.

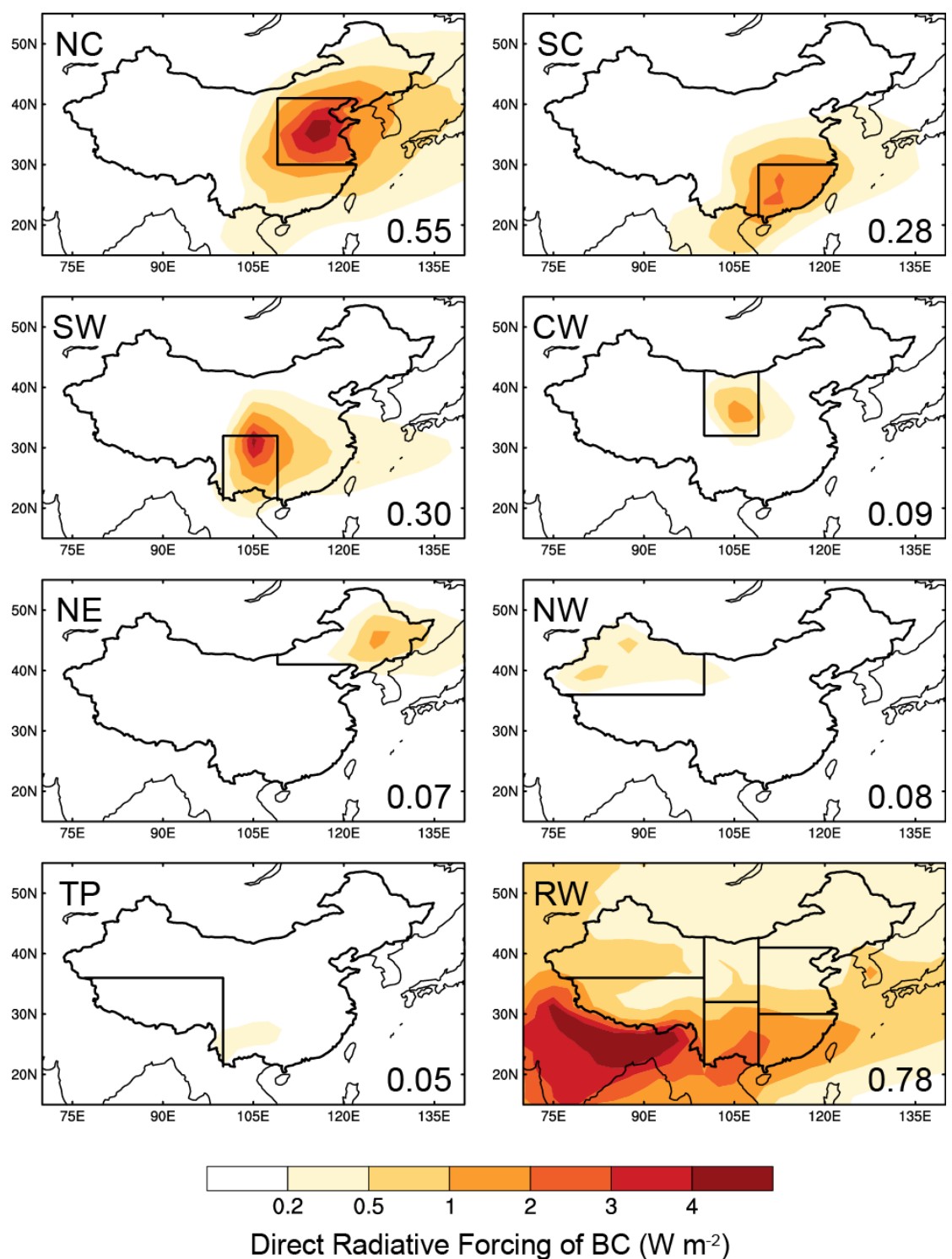

**Figure 11.** Spatial distribution of annual mean direct radiative forcing (DRF) of BC (W m$^{-2}$) at the top of the atmosphere originating from the tagged BC source regions in China (NC, SC, SW, CW, NE, NW, and TP) and source outside China (RW). Regionally averaged forcing in China contributed by individual source regions is shown at the bottom right of each panel.

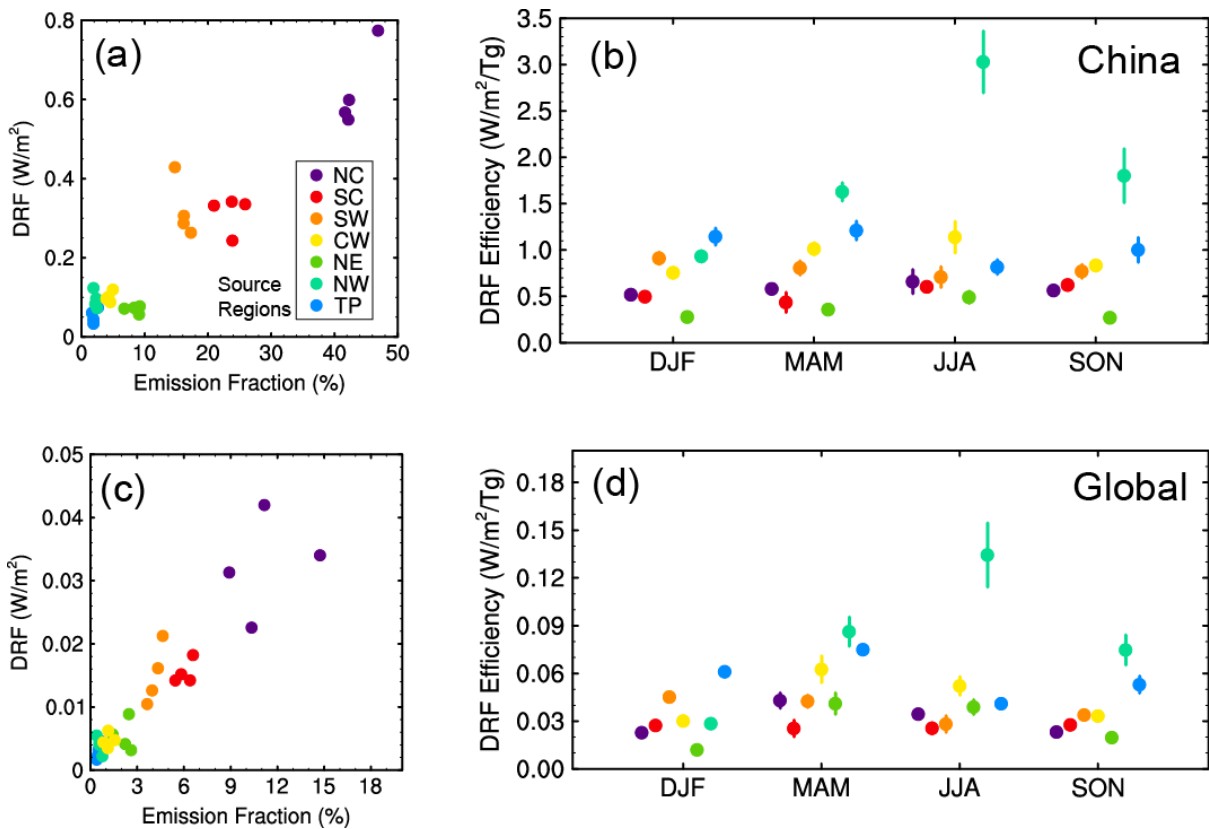

1377

1378

**Figure 12.** (a, c) BC seasonal DRF averaged over China as a function of BC

emission fraction (the ratio of regional emission to the total emission over China and

global, respectively, unit: %) for each of the tagged regions. (b, d) Seasonal DRF

efficiency of BC (W m$^{-2}$ Tg$^{-1}$) for each of the tagged source regions over China and

globally, respectively. The efficiency is defined as the DRF divided by the

corresponding scaled annual emission (seasonal emission multiplied by 4). Error bars

indicate 1-σ of mean values during years 2010–2014.


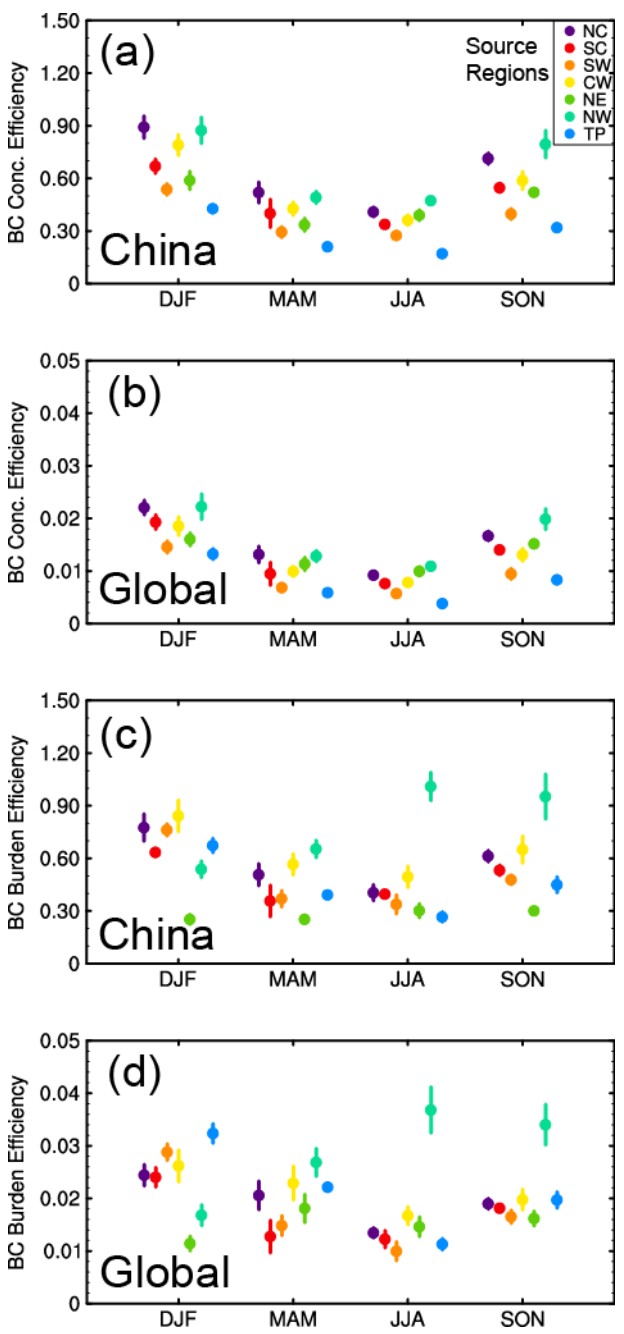



**Figure 13.** Seasonal (a, b) near-surface concentration ($\mu g\ m^{-3}\ Tg^{-1}$) and (c, d) column
burden ($mg\ m^{-2}\ Tg^{-1}$) efficiency of BC for each of the tagged source regions over
China and globally, respectively.