# Peer review of "1. Introduction"

_Atmospheric Chemistry and Physics, 2016_

## Referee Comment (RC1) · Anonymous Referee #1 · 19 Dec 2016

The authors used the CESM model with a source-tagging method to quantify the source attributions for BC direct radiative forcing (DRF) and concentration as well as polluted events. They found that in addition to regional emissions within China, emissions outside China also contribute to a large portion of BC DRF over China. This study could improve the understanding on BC pollution in China and provide implications for policy makers. Before this manuscript can be considered for publication, I have a few comments that need to be addressed by the authors.

1. One critical factor influencing BC direct radiative effect is its optical properties (absorption and extinction cross section, asymmetry factor, and single scattering albedo). Recent studies (e.g., He et al. 2015, 2016b) showed that BC optical properties vary significantly (by up to more than a factor of two) due to different coating structures and aging stages during BC aging process, which further affects direct radiative effect. (1)

Could the authors add some discussions on this aspect with reference to these recent studies, for example, potential uncertainty in their results caused by this factor? (2) Could this variation in BC optical properties due to coating structures contribute to the model biases in BC AAOD simulations as discussed in the first paragraph of Section 3.2? (3) It would be helpful if the authors could add more details on how the MAM3 model computes BC optical properties. For example, does it assume a core-shell structure for internally mixed BC?

References:

He, C., et al. : Variation of the radiative properties during black carbon aging: theoretical and experimental intercomparison, Atmos. Chem. Phys., 15, 11967-11980, doi:10.5194/acp-15-11967-2015, 2015.

He, C., et al. : Intercomparison of the GOS approach, superposition T-matrix method, and laboratory measurements for black carbon optical properties during aging, J. Quant. Spectrosc. Radiat. Transf., 184, 287–296, doi:10.1016/j.jqsrt.2016.08.004, 2016b.

2. For BC emissions, a number of global and regional emission inventories have been developed, which showed large uncertainties and differences among each other (e.g., Fig. 4 in Wang et al., 2014). It would be helpful if the authors could discuss the uncertainty associated with the emission inventory used in this study and how this inventory compares with previous ones for both inside and outside China, since the authors pointed out that emissions outside China also contribute a lot to BC DRF in China.

Reference:

Wang, R., et al. : Trend in Global Black Carbon Emissions from 1960 to 2007, Environ. Sci. Technol., 48, 6780$-$6787, doi: 10.1021/es5021422, 2014.

3. Another important factor affecting BC simulations is aging process, which directly

alters BC wet scavenging and lifetime. As pointed out by some recent studies (e.g., Oshima et al., 2009; He et al., 2016a), applying microphysical BC aging schemes could significantly improve simulations of BC concentrations compared with simplified aging parameterizations. (1) Could the authors briefly describe how BC aging is treated/computed in their model? (2) It would be helpful if the authors could briefly discuss the BC aging effect on their results with reference to these recent studies. (3) The authors mentioned in Lines 255–256 that model biases in BC concentration over China is likely due to inaccurate emissions and wet scavenging. Could this bias also be caused by model uncertainty related to BC aging? Some discussions would be useful.

Reference:

He, C., Li, Q., Liou, K.-N., Qi, L., Tao, S., and Schwarz, J. P.: Microphysics-based black carbon aging in a global CTM: constraints from HIPPO observations and implications for global black carbon budget, Atmos. Chem. Phys., 16, 3077-3098, doi:10.5194/acp-16-3077-2016, 2016a.

Oshima, N., et al. : Aging of black carbon in outflow from anthropogenic sources using a mixing state resolved model: Model development and evaluation, J. Geophys. Res., 114, D06210, doi:10.1029/2008JD010680, 2009.

4. The authors derived BC AAOD from AERONET observations by using the method in Bond et al. (2013). However, a recent study by Schuster et al. (2016) pointed out some weaknesses and problems related to the Bond et al. (2013) method. Could the author briefly discuss this issue? How would this affect the results in this study?

Reference:

Schuster, G. L., et al. : Remote sensing of soot carbon – Part 2: Understanding the absorption Ångström exponent, Atmos. Chem. Phys., Âǎ16, 1587–1602, Âǎdoi:10.5194/acp-16-1587-2016, 2016.

---

## Referee Comment (RC2) · Anonymous Referee #2 · 5 Jan 2017

Review of "Source attribution of black carbon and its direct radiative forcing in China" by Yang et al

Yang et al. investigated the BC source attributions (or more specifically, region attributions) in China with a source-tagging technique by employing a global climate model, NCAR's Community Earth System Model. They found out that BC emissions from local (inside China) and non-local (outside China) are both generally important contributions to air quality in different regions of China, BC outflow from East Asia and direct radiative forcings. Overall, this paper is a helpful addition to our community that attempts to improve our understanding of BC source-receptor relationship. This paper generally reads well and is within the scope of ACP. However, before it can be accepted for publication in ACP, I have several comments that need to be properly addressed.

Major comments:

An important part of this study was to quantify the BC source contributions to trans-boundary and trans-pacific transport. In terms of model performance evaluation, this study only validated model simulations with observed BC surface concentrations from CAWNET and AAOD from AERONET over China. We don't know the efficiency of BC outflow from East Asia. In this paper, it obviously missed the evaluations of model simulated vertical profiles of BC against aircraft campaign observations, e.g. A-FORCE and HIPPO, which should be employed to compare with model simulations.

Other minor comments:

Line 124: the reference Hoesly et al., 2016 is missing in the reference list.

Line 162: A brief description of dry/wet deposition scheme for BC in CAM is lacking here, especially the wet scavenging and how it is improved following H. Wang et al. (2013).

Line 334-336: This sentence should be corrected as "AI derived from Total Ozone Mapping Spectrometer (TOMS) measurements also shows similar pattern as simulate AAOD (Fig. S2)."

Line 339-344: What's the assumption here? Why the ratio of AAOD to AI should be the same between western and eastern China? What is the role of dust here in assisting the speculation?

Line 424-426: I think BC emissions from SC are also important for the column burdens over continental China in some seasons (e.g. JJA and SON), which needs to be outlined as well.

Line 443: "Figure S4a" should be replaced with "Fig. S5a".

Line 443-463: It is helpful for the authors to make supplemental plots showing the anomalous winds during polluted days that favor the accumulation of pollutants over each region.

Line 505-509: Why the authors only choose the latitude range along longitude 150°E, not a domain covering East China Sea and West Pacific to quantitatively assess the BC contributions from China and outside China, similar to that impact over West United States?

Line 509-510: I get lost here. It is not clear to me that 58% contribution from China emissions is for outflow or something else. Authors need to clarify this.

Line 531-538: I think the authors should list a table to compare your results with other studies, including annual BC emission budgets, burden, lifetime, DRF and DRF efficiency.

Line 654-655: Other modeling studies also found model low bias over China using CAWNET, e.g. Huang et al., 2013; Wang et al., 2014, which can be referenced here.

Line 669: "and" is missing between "modeled" and "observed".

Reference:

Huang, Y., S. Wu, M. K. Dubey, and N. H. F. French, Impact of aging mechanism on model simulated carbonaceous aerosols, Atmos. Chem. Phys., 13, 6329–6343, doi:10.5194/acp-13-6329-2013, 2013.

Wang, Q., D.J. Jacob, J.R Spackman, A.E. Perring, J.P. Schwarz, N. Moteki, E.A. Marais, C. Ge, J. Wang and S.R.H. Barrett, Global budget and radiative forcing of black carbon aerosol: constraints from pole-to-pole (HIPPO) observations across the Pacific, J. Geophys. Res., 119, 195-206, 2014.

---

## Referee Comment (RC3) · Anonymous Referee #3 · 18 Jan 2017

This study quantified source contributions of black carbon (BC) mass concentrations, trans-Pacific transport of BC, and direct radiative forcing of BC from seven regions in China using the Community Earth System Model with a source-tagging technique. The authors showed that BC concentrations were dominated by local emissions for regions with high emissions (e.g., North China, South China), whereas non-local emissions were important for regions with low emissions (e.g., Northwest China, Tibetan Plateau). They also showed that emissions from China and other regions were equally important for the BC outflow from East Asia and that emissions from China would be important for air quality in western United States. The annual mean direct radiative forcing of BC in China in their simulations was 2.3 W m-2, and the contribution from emissions in China was estimated to be 66%.

The purpose of this study is interesting and the results obtained in this study are important to understand BC behavior in the atmosphere over East Asia. I think the authors should describe several points (shown below) more clearly, but overall the manuscript is written well and is suitable for the publication of this journal.

Major comments:

(1) Importance of BC in air quality problems

The authors sometimes use BC as an indicator of pollution (or air quality) in China (Lines 39-40, Lines 101-102, Lines 429-431, and Lines 571-572). However, I think it is questionable whether the concentrations and/or source contributions of BC can be used to represent those of total aerosols. Inorganic and organic species are dominant in China during polluted days, and spatial/temporal variations and source contributions of these species are largely different from those of BC because spatial distributions of emissions (e.g., BC v.s. SO2) and formation processes (primary v.s. secondary) are considerably different. For example, Matsui et al. (2009) showed that primary aerosols around Beijing were controlled by emissions within 100 km around Beijing within the preceding 24 h, while emissions as far as 500 km and within the preceding 3 days were found to affect secondary aerosols. Therefore, it is not always correct to extend the results of BC (e.g., source contributions) to the discussions on pollution and air quality because inorganic and organic species could have larger contributions from non-local emissions than BC. Please consider this point and describe the limitation of using BC results only in the discussions of air quality problems. In addition, please show the percentage of BC mass to total mass (PM2.5) in China in the manuscript.

(2) Treatment of optical property and CCN activity of BC (Lines 151-169)

I could not find the description on the treatment of optical property (well-mixed, core-shell, or others) and CCN activity (conversion from hydrophobic to hydrophilic BC) of BC in the MAM3 model. I assume that well-mixed optical treatment is used to calculate BC absorption and that all BC particles are treated as hydrophilic BC in MAM3. Please describe the treatment of optical property and CCN activity of BC in the manuscript,

and add some description on the potential impact (uncertainty) of these treatments on the estimation of BC concentrations, trans-Pacific transport of BC, AAOD, and direct radiative forcing of BC and their source contributions.

Other comments:

(3) Line 70

Please describe the reason of the faster regional removal.

(4) Lines 168-169

Please clarify the definition of the direct radiative forcing of BC. Is this calculated from the difference of two radiative transfer calculations with and without BC for the clear-sky condition?

(5) Lines 182-204

Please show the difference of BC emission fluxes between the emission inventory used in this study and other emission inventories (e.g., INTEX-B, HTAP). The values are shown later (at Lines 534-538), but I think it is better to show them here. In addition, please add some comments on the impact of larger values of BC emissions in this study on the estimation of source contributions of BC. Can you add the values of BC emissions from outside China (e.g., India, Southeast Asia, Japan, Korea) to Figure 1b?

(6) Lines 261-263

You can show the contributions from outside China quantitatively from the tagged simulation results.

(7) Line 281

Please describe the reason of BC underestimation by up to a factor of 20.

(8) Line 343

I cannot find large sources of BC in Northwest China in Figure 1a. Does the description

here mean that there may be large sources of BC which are not considered in the emission inventory? Can you show the contribution of BC and dust to AAOD (in model) over this region? I think dust is dominant over this region.

(9) Lines 667-669

Related to the comment (2), is an internally-mixed treatment used in the calculations of AAOD? If so, AAOD should be lower (underestimated more) when more realistic BC mixing state treatment is used in the optical calculations.

Reference:

Matsui, H., et al. (2009), Spatial and temporal variations of aerosols around Beijing in summer 2006: Model evaluation and source apportionment, J. Geophys. Res., 114, D00G13, doi:10.1029/2008JD010906.

---

## Author Comment (AC1) · 2 Feb 2017

**Responses to Reviewer #1**

The authors used the CESM model with a source-tagging method to quantify the source attributions for BC direct radiative forcing (DRF) and concentration as well as polluted events. They found that in addition to regional emissions within China, emissions outside China also contribute to a large portion of BC DRF over China. This study could improve the understanding on BC pollution in China and provide implications for policy makers. Before this manuscript can be considered for publication, I have a few comments that need to be addressed by the authors.

1. One critical factor influencing BC direct radiative effect is its optical properties (absorption and extinction cross section, asymmetry factor, and single scattering albedo). Recent studies (e.g., He et al. 2015, 2016b) showed that BC optical properties vary significantly (by up to more than a factor of two) due to different coating structures and aging stages during BC aging process, which further affects direct radiative effect. (1) Could the authors add some discussions on this aspect with reference to these recent studies, for example, potential uncertainty in their results caused by this factor? (2) Could this variation in BC optical properties due to coating structures contribute to the model biases in BC AAOD simulations as discussed in the first paragraph of Section 3.2? (3) It would be helpful if the authors could add more details on how the MAM3 model computes BC optical properties. For example, does it assume a core-shell structure for internally mixed BC?
References:
He, C., et al. : Variation of the radiative properties during black carbon aging: theoretical and experimental intercomparison, Atmos. Chem. Phys., 15, 11967-11980, doi:10.5194/acp-15-11967-2015, 2015.
He, C., et al. : Intercomparison of the GOS approach, superposition T-matrix method, and laboratory measurements for black carbon optical properties during aging, J. Quant. Spectrosc. Radiat. Transf., 184, 287–296, doi:10.1016/j.jqsrt.2016.08.004, 2016b.

Response:
   (1) Thanks for the suggestion. We have added discussions of the influence of aging processes on BC optical properties to the Conclusions and Discussions section, along with the references provided by the referee, as "BC aging in the atmosphere is important for BC concentration and its optical properties, which transforms BC from hydrophobic aggregates to hydrophilic particles coated with soluble materials. He et al. (2015, 2016a) found that BC optical properties varied by a factor of two or more due to different coating structures during BC aging process based on their theoretical and

experimental intercomparison. Oshima et al. (2009) and He et al. (2016b) pointed out that the use of various microphysical BC aging schemes could significantly improve simulations of BC concentrations compared to the simplified aging parameterizations. Liu et al. (2012) also reported that the wet removal rate of BC simulated in standard CAM5 is 60% higher than AeroCom multi-model mean due to the rapid or instantaneous aging of BC. H. Wang et al. (2013) showed that the explicit treatment of BC aging process with slow aging assumptions in CAM5 could significantly increase BC lifetime and the efficiency of BC long-range transport. In the three-mode aerosol module (MAM3) of CAM5 used in this study, the aging process of BC is neglected by assuming the immediate internal mixing of BC with other aerosol species in the same mode. This assumption could lead to an overestimation of wet removal of BC and, therefore, an underestimation of BC concentrations, absorption optical depth (Fig. 3) and direct radiative forcing. In addition, the internally-mixed optical treatment in CAM5 could also cause bias in BC absorption calculation. However, H. Wang et al. (2014) examined source-receptor relationships for BC under the different BC aging assumptions and found that the quantitative source attributions varied slightly while the qualitative source-receptor relationships still hold. Therefore, although the magnitude of simulated BC and its optical properties could be underestimated due to the instantaneous aging of BC and uncertainty in coating structures, we expect that the aging treatment in MAM3 of CAM5 should not influence the qualitative source attributions examined in this study."

(2) We agree that uncertainties in BC optical properties due to coating structures and/or aging could contribute to model biases in simulated BC concentrations and AAOD. We have added this statement in the discussion section. Please see the revisions above.

(3) In the MAM3 aerosol module of CAM5 used in this study, the aging process of BC is neglected by assuming an instantaneous mixing of BC with other aerosol species in the same accumulation mode, which has been added in the discussion section. We also add a more detailed description of the calculation of aerosol optical properties in the Methods section, as "Aerosol optical properties for each mode are parameterized according to Ghan and Zaveri (2007). Refractive indices for aerosols are taken from the OPAC (optical properties for aerosols and clouds) software package (Koepke and Schult, 1998), but for BC at solar wavelengths the values are updated from Bond and Bergstrom (2006)."

2. For BC emissions, a number of global and regional emission inventories have been developed, which showed large uncertainties and differences among each other (e.g., Fig. 4 in Wang et al., 2014). It would be helpful if the authors could discuss the uncertainty associated with the emission inventory used in this study and how this inventory compares with previous ones for both inside and outside China, since the authors pointed out that emissions outside

China also contribute a lot to BC DRF in China.

Reference:
Wang, R., et al. : Trend in Global Black Carbon Emissions from 1960 to 2007, Environ. Sci. Technol., 48, 6780−6787, doi: 10.1021/es5021422, 2014.

Response:
Uncertainty in China BC emissions has been estimated as –43% to 93% by Lu et al. (2011), –50% to 164% by Qin and Xie (2012), ±176% by Kurokawa et al. (2013), and –28 to 126% by Zhao et al. (2013). The BC emissions estimates used here for China in 2010 are 40% higher than those of Zhao et al. (2013) and Lu et al. (2011) and 30% higher than Klimont et al. (2016), in large part due to a higher estimate of BC emissions from coal coke production. Emissions from coke production are particularly uncertain given that "there are no measurements for $PM_{2.5}$ and BC emissions" (Huo et al. 2012) available to guide inventory estimates. Total rest of the world emissions other than China, which appear to be a major contributor to burdens over western regions, are within 1% of those from Klimont et al. (2016). We have added these discussions in Conclusions and discussions section.

We have added Table S1 to compare the anthropogenic emissions used in this study with emissions from some previous studies. The anthropogenic emissions of BC in China in 2010–2014 are larger than those used in the previous studies for earlier years, partly as a result of a higher estimate of BC emissions from coal coking production. The higher emissions likely lead to higher concentrations and direct radiative forcing, and source contributions of BC in China, compared to the values reported in these studies. We have added these descriptions in the Methods section.

We also revised Fig. 1 to include BC emissions from outside China. Emissions at regional scale are summarized here instead of Country level because the model revolution is a bit course to characterize emissions by countries. Total BC emissions from neighboring regions including rest of East Asia (REA, with China excluded), South Asia (SAS), Southeast Asia (SEA), and Russia/Belarussia/Ukraine (RBU) are shown in Figure 1c. These source regions outside China are consistent with source regions defined in the second phase of Hemispheric Transport of Air Pollution (HTAP2). South Asia and Southeast Asia have relatively high emissions. They may dominate the contribution to concentrations and direct radiative forcing of BC in China, especially southern and western China, from foreign sources through long-range transport. We have also added these in the Methods section. We did not compare the emissions outside China with other studies, which is beyond the scope of this study. However, the comparison of CEDS emissions with other emission inventories can be found in Hoesly et al. (2017), which includes detailed information of the CEDS emissions and will be submit very soon.

**Table S1.** Comparison of CEDS annual mean anthropogenic BC emissions in China with those used in other studies

| | Year | Anthropogenic emission in China (Gg/yr) |
|---|---|---|
| CEDS (Hoesly et al., 2016; this study) | 2010–2014 | 2467 |
| MIX (Li et al., 2017) | 2010 | 1765 |
| HTAP V2.2 (Janssens- Maenhout et al., 2015) | 2010 | 1741 |
| Lu et al. (2011) | 2010 | 1751 |
| Qin and Xie (2012) | 2009 | 1764 |
| Wang et al. (2012) | 2007 | 1879 |
| INTEX-B (Zhang et al., 2009) | 2006 | 1811 |

[Figure]

**Figure 1.** (a) Spatial distribution of annual mean total emissions (anthropogenic plus biomass burning, units: g C m$^{-2}$ yr$^{-1}$) of black carbon (BC) averaged over 2010–2014. The geographical BC source regions are selected as North China (NC, 109°E–east boundary, 30°–41°N), South China (SC, 109°E–east boundary, south boundary–30°N), Southwest China (SW, 100°–109°N, south boundary–32°N), Central-West China (CW, 100°–109°N, 32°N–north boundary), Northeast China (NE, 109°E–east boundary, 41°N–north boundary), Northwest China (NW, west boundary–100°E, 36°N–north boundary), and Tibetan Plateau (TP, west boundary–100°E, south boundary–36°N) in China and regions outside of China (RW, rest of the world). (b) Seasonal mean total emissions (units: Gg C, Gg = 10$^9$g) of BC from the seven BC source regions in China and (c) emissions from rest of East Asia (REA, with China excluded), South Asia (SAS), Southeast Asia (SEA), and Russia/Belarussia/Ukraine (RBU).

3. Another important factor affecting BC simulations is aging process, which directly alters BC wet scavenging and lifetime. As pointed out by some recent studies (e.g., Oshima et al., 2009; He et al., 2016a), applying microphysical BC aging schemes could significantly improve simulations of BC concentrations compared with simplified aging parameterizations. (1) Could the authors briefly describe how BC aging is treated/computed in their model? (2) It would be helpful if the authors could briefly discuss the BC aging effect on their results with reference to these recent studies. (3) The authors mentioned in Lines 255–256 that model biases in BC concentration over China is likely due to inaccurate emissions and wet scavenging. Could this bias also be caused by model uncertainty related to BC aging? Some discussions would be useful.

reference:

He, C., Li, Q., Liou, K.-N., Qi, L., Tao, S., and Schwarz, J. P.: Microphysics-based black carbon aging in a global CTM: constraints from HIPPO observations and implications for global black carbon budget, Atmos. Chem. Phys., 16, 3077-3098, doi:10.5194/acp-16-3077-2016, 2016a.

Oshima, N., et al. : Aging of black carbon in outflow from anthropogenic sources using a mixing state resolved model: Model development and evaluation, J. Geophys. Res., 114, D06210, doi:10.1029/2008JD010680, 2009.

Response:
    (1) Thanks for the suggestion. We had now added more description of the BC aging treatment and related discussions to the paper. Please see the response to comment #1 above.
    (2) Following the referee's suggestion, we had now added more discussions in this regard. Please see the response to comment #1 above.
    (3) Yes, the overestimation of wet scavenging is partly caused by the assumption of instantaneous aging of BC in the model. We have added the message to the manuscript, as "Larger wet removal rate and shorter lifetime of aerosols along with the instantaneous aging of BC in the MAM3 can also lead to the lower concentrations of BC (e.g., Wang et al., 2011; Liu et al., 2012; H. Wang et al., 2013; Kristiansen et al., 2016)." and in discussion section as "This assumption could lead to an overestimation of wet removal of BC and, therefore, an underestimation of BC concentrations, absorption optical depth (Fig. 3) and direct radiative forcing."

4. The authors derived BC AAOD from AERONET observations by using the method in Bond et al. (2013). However, a recent study by Schuster et al. (2016)

pointed out some weaknesses and problems related to the Bond et al. (2013) method. Could the author briefly discuss this issue? How would this affect the results in this study?

Reference:
Schuster, G. L., et al. : Remote sensing of soot carbon – Part 2: Understanding the absorption Ångström exponent, Atmos. Chem. Phys., 16, 1587–1602, doi:10.5194/acp-16-1587-2016, 2016.

Response:
    We have included a discussion of this caveat associated with the BC AAOD comparison, as "Note that, the observed AAOD of BC is derived from AERONET measurements using the absorption Ångström exponent. A recent study (Schuster et al., 2016) reported that absorption Ångström exponent is not a robust parameter for separating out carbonaceous absorption in the AERONET database, which could cause biases in the AAOD estimates."

References:

He, C., Liou, K.-N., Takano, Y., Zhang, R., Levy Zamora, M., Yang, P., Li, Q., and Leung, L. R.: Variation of the radiative properties during black carbon aging: theoretical and experimental intercomparison, Atmos. Chem. Phys., 15, 11967-11980, doi:10.5194/acp-15-11967-2015, 2015.

He, C., Takano, Y., Liou, K.-N., Yang, P., Li, Q., and Mackowski, D. W.: Intercomparison of the GOS approach, superposition T- matrix method, and laboratory measurements for black carbon optical properties during aging, J. Quant. Spectrosc. Ra., 184, 287–296, doi:10.1016/j.jqsrt.2016.08.004, 2016a.

Kristiansen, N. I., Stohl, A., Olivié, D. J. L., Croft, B., Søvde, O. A., Klein, H., Christoudias, T., Kunkel, D., Leadbetter, S. J., Lee, Y. H., Zhang, K., Tsigaridis, K., Bergman, T., Evangeliou, N., Wang, H., Ma, P.-L., Easter, R. C., Rasch, P. J., Liu, X., Pitari, G., Di Genova, G., Zhao, S. Y., Balkanski, Y., Bauer, S. E., Faluvegi, G. S., Kokkola, H., Martin, R. V., Pierce, J. R., Schulz, M., Shindell, D., Tost, H., and Zhang, H.: Evaluation of observed and modelled aerosol lifetimes using radioactive tracers of opportunity and an ensemble of 19 global models, Atmos. Chem. Phys., 16, 3525-3561, doi:10.5194/acp-16-3525-2016, 2016.

Liu, X., et al. (2012), Toward a minimal representation of aerosols in climate models: Description and evaluation in the Community Atmosphere Model CAM5, Geosci. Model Dev., 5, 709–739, doi:10.5194/gmd-5-709-2012.

Liu, X., Ma, P.-L., Wang, H., Tilmes, S., Singh, B., Easter, R. C., Ghan, S. J., and Rasch, P. J.: Description and evaluation of a new four-mode version of the Modal Aerosol Module (MAM4) within version 5.3 of the Community Atmosphere Model, Geosci. Model Dev., 9, 505-522, doi:10.5194/gmd-9-505-2016, 2016.

Wang, H., R. C. Easter, P. J. Rasch, M. Wang, X. Liu, S. J. Ghan, Y. Qian, J.-H. Yoon, P.-L. Ma, and V. Vinoj (2013), Sensitivity of remote aerosol distributions to representation of cloud-aerosol interactions in a global climate model, Geosci. Model Dev., 6, 765–782, doi:10.5194/gmd-6-765-2013.

Wang, H., P. J. Rasch, R. C. Easter, B. Singh, R. Zhang, P.-L. Ma, Y. Qian, S. J. Ghan, and N. Beagley (2014), Using an explicit emission tagging method in global modeling of source-receptor relationships for black carbon in the Arctic: Variations, sources, and transport pathways, J. Geophys. Res. Atmos., 119, 12,888–12,909, doi:10.1002/ 2014JD022297.

Ghan, S. J., and R. A. Zaveri (2007), Parameterization of optical properties for hydrated internally mixed aerosol, J. Geophys. Res., 112, D10201, doi:10.1029/2006JD007927.

Bond, T. C., and R. W. Bergstrom, Light absorption by carbonaceous particles: An investigative review, Aerosol. Sci. Technol., 40, 27–67, doi:10.1080/02786820500421521, 2006.

Koepke, M. H. P., and I. Schult, Optical properties of aerosols and clouds: The software package opac, Bull. Am. Meteorol. Soc., 79, 831–844, 1998, doi:10.1175/1520-0477(1998)079<0831:OPOAAC>2.0.CO;2.

Wang, M., S. Ghan, M. Ovchinnikov, X. Liu, R. Easter, E. Kassianov, Y. Qian, and H. Morrison (2011), Aerosol indirect effects in a multi-scale aerosol-climate model PNNL-MMF, Atmos. Chem. Phys., 11, 5431–5455, doi:10.5194/acp-11-5431-2011.

He, C., Li, Q., Liou, K.-N., Qi, L., Tao, S., and Schwarz, J. P.: Microphysics-based black carbon aging in a global CTM: constraints from HIPPO observations and implications for global black carbon budget, Atmos. Chem. Phys., 16, 3077-3098, doi:10.5194/acp-16-3077-2016, 2016b.

Oshima, N., M. Koike, Y. Zhang, Y. Kondo, N. Moteki, N. Takegawa, and Y. Miyazaki (2009), Aging of black carbon in outflow from anthropogenic

sources using a mixing state resolved model: Model development and evaluation, J. Geophys. Res., 114, D06210, doi:10.1029/2008JD010680.

Schuster, G. L., Dubovik, O., Arola, A., Eck, T. F., and Holben, B. N.: Remote sensing of soot carbon – Part 2: Understanding the absorption Ångström exponent, Atmos. Chem. Phys., 16, 1587-1602, doi:10.5194/acp-16-1587-2016, 2016.

Kurokawa, J., Ohara, T., Morikawa, T., Hanayama, S., Janssens-Maenhout, G., Fukui, T., Kawashima, K. and Akimoto, H.: Emissions of air pollutants and greenhouse gases over Asian regions during 2000–2008: Regional Emission inventory in ASia (REAS) version 2, Atmospheric Chem. Phys., 13(21), 11019–11058, 2013.

Qin, Y. and Xie, S. D.: Spatial and temporal variation of anthropogenic black carbon emissions in China for the period 1980–2009, Atmos Chem Phys, 12(11), 4825–4841, doi:10.5194/acp-12-4825-2012, 2012.

Zhao, Y., Zhang, J. & Nielsen, C. P. 2013. The effects of recent control policies on trends in emissions of anthropogenic atmospheric pollutants and $CO_2$ in China. Atmos. Chem. Phys., 13, 487-508.

Lu, Z., Zhang, Q. & Streets, D. G. 2011. Sulfur dioxide and primary carbonaceous aerosol emissions in China and India, 1996–2010. Atmos. Chem. Phys., 11, 9839-9864.

Huo, H., Lei, Y., Zhang, Q., Zhao, L. & He, K. 2012. China's coke industry: Recent policies, technology shift, and implication for energy and the environment. Energy Policy, 51, 397-404.

Klimont, Z., Kupiainen, K., Heyes, C., Purohit, P., Cofala, J., Rafaj, P., Borken-Kleefeld, J. and Schöpp, W.: Global anthropogenic emissions of particulate matter including black carbon, Atmospheric Chem. Phys. Discuss., 1–72, doi:10.5194/acp-2016-880, 2016.

---

## Author Comment (AC2) · 2 Feb 2017

**Responses to Reviewer #2**

Yang et al. investigated the BC source attributions (or more specifically, region attributions) in China with a source-tagging technique by employing a global climate model, NCAR's Community Earth System Model. They found out that BC emissions from local (inside China) and non-local (outside China) are both generally important contributions to air quality in different regions of China, BC outflow from East Asia and direct radiative forcings. Overall, this paper is a helpful addition to our community that attempts to improve our understanding of BC source-receptor relationship. This paper generally reads well and is within the scope of ACP. However, before it can be accepted for publication in ACP, I have several comments that need to be properly addressed.

Major comments:

An important part of this study was to quantify the BC source contributions to trans-boundary and trans-pacific transport. In terms of model performance evaluation, this study only validated model simulations with observed BC surface concentrations from CAWNET and AAOD from AERONET over China. We don't know the efficiency of BC outflow from East Asia. In this paper, it obviously missed the evaluations of model simulated vertical profiles of BC against aircraft campaign observations, e.g. A-FORCE and HIPPO, which should be employed to compare with model simulations.

Response:
    Thanks for the suggestion. The simulated BC vertical profile in CAM5 has been extensively evaluated in many previous studies. Liu et al. (2012) compared the observed and simulated BC vertical profiles in the tropics, middle latitudes, and high latitudes from six aircraft campaigns: AVE Houston (NASA Houston Aura Validation Experiment), CR-AVE (NASA Costa Rica Aura Validation Experiment), TC4 (Tropical Composition, Cloud and Climate Coupling), CARB (NASA initiative in collaboration with California Air Resources Board), ARCTAS (NASA Arctic Research of the Composition of the Troposphere from Aircraft and Satellite), and ARCPAC (NOAA Aerosol, Radiation, and Cloud Processes affecting Arctic Climate), as well as BC vertical profiles over the Arctic Ocean and the remote Pacific Ocean during the HIPPO (HIAPER Pole-to-Pole Observations) campaign. They found that measured BC mixing ratios showed a strong gradient from the boundary layer to the free troposphere in the tropics, while modeled BC mixing ratios showed a smaller decrease with altitude in the free troposphere, thus overestimating observations above 500 hPa. Compared to HIPPO campaign, the CAM5 model captured the vertical variations of BC mixing ratio reasonably well in the

SH high latitudes and NH and SH mid-latitudes. However, modeled BC showed less vertical reduction in the tropics, thus significantly overestimating BC in the upper troposphere.

Wang et al. (2013) implemented in CAM5 a unified treatment of wet removal and vertical transport of aerosols by convection, which included an explicit secondary activation of aerosols being laterally entrained into convective clouds above cloud base. The comparisons between the new CAM5 simulated vertical profiles of BC mass mixing ratios and the HIPPO and the field campaign aircraft observations showed a substantial improvement in the simulation of BC in mid- and upper troposphere, where the excessive BC was significantly reduced. All of these key model improvements by Wang et al. (2013) have been included in the version of CAM5 being used in the present study. Therefore, we did not duplicate the evaluation of BC vertical profiles with HIPPO observation. We have now revised the description before model evaluation to make it clear, as "The simulations of aerosols, especially BC, using CAM5 have been extensively evaluated against observations including aerosol mass and number concentrations, vertical profiles, aerosol optical properties, aerosol deposition, and cloud-nucleating properties in several previous studies (e.g., Liu et al., 2012, 2016; H. Wang et al., 2013; Ma et al., 2013b; Jiao et al., 2014; Qian et al., 2014; R. Zhang et al., 2015a,b)."

In addition, as the referee suggested, we have added a comparison of the simulated BC vertical profile with A-FORCE measurements over East Asia (see Figure S2). The model successfully reproduces the vertical profile of BC. The bias is relatively small. We have also added a relevant discussion to the revised manuscript, as "Figure S2 compares the observed and simulated vertical profiles of BC concentrations in the East-Asian outflow region. The model successfully reproduces the vertical profile of BC that was measured in March–April 2009 during the A-FORCE field campaign and reported by Oshima et al. (2012)."

[Figure]

Figure S2. Observed and simulated mean vertical profiles of BC concentrations in the East-Asian outflow region. The observed BC profile is from the A-FORCE field campaign conducted over the Yellow Sea, the East China Sea, and the western Pacific Ocean in March–April 2009 (Oshima et al., 2012).

Other minor comments:

Line 124: the reference Hoesly et al., 2016 is missing in the reference list.
Response:
    The paper is still in preparation. A draft can be made available to referees upon request.

Line 162: A brief description of dry/wet deposition scheme for BC in CAM is lacking here, especially the wet scavenging and how it is improved following H. Wang et al. (2013).
Response:
    Aerosol dry deposition velocities are calculated using the Zhang et al. (2001) parameterization. The wet deposition of aerosols in our CAM5 model includes in-cloud wet removal (i.e., activation of interstitial aerosols to cloud-borne particles followed by precipitation scavenging) and below-cloud wet removal (i.e., capture of interstitial aerosol particles by falling precipitation particles) for both stratiform and convective clouds. Aerosol activation is calculated with the parameterization of Abdul-Razzak and Ghan (2000) for stratiform cloud throughout the column and convective cloud at cloud base, while the secondary activation above convective cloud base has a simpler treatment with an assumed maximum supersaturation in convective updrafts (H. Wang et al., 2013). The unified treatment for convective transport and aerosol wet removal along with the explicit aerosol activation above convective cloud base was developed by H. Wang et al. (2013) and included in the CAM5 version being used in this study. As discussed in the response to the major comment, this implementation reduces the excessive BC aloft and better simulates observed BC concentrations in the mid- to upper-troposphere.
    We have now added these descriptions to the Methods section.

Line 334-336: This sentence should be corrected as "AI derived from Total Ozone Mapping Spectrometer (TOMS) measurements also shows similar pattern as simulate AAOD (Fig. S2)."
Response:
    Corrected.

Line 339-344: What's the assumption here? Why the ratio of AAOD to AI should be the same between western and eastern China? What is the role of dust here in assisting the speculation?

Response:

Based on the comparison of AAOD between the model simulation and AERONET, we found that the model reproduced well the observed AAOD over eastern China. Therefore, we assume that the ratio of modeled AAOD and satellite AI (indicator for absorbing aerosols) is correct over eastern China. The AAOD/AI over eastern China is much larger than western China, suggesting that the ratio AAOD/AI is lower than the true value and AAOD or BC burden is likely underestimated in the model. Both AAOD and AI represent absorbing aerosols, the ratio of AAOD to AI may be different but similar between different regions in China. The difference in the ratio between eastern and western China is quite large, suggesting the existence of a significant bias. This is consistent with the contrast of biases in near-surface BC concentrations between the two regions (shown in Fig. 3). However, both BC and dust can contribute to AAOD and AI. Potential biases in the modeled dust could also lead to the inconsistence of AAOD/AI between eastern and western China.

We have revised the description to make it clear, as "If we assume that the simulated AAOD do not have large biases over eastern China based on the evaluation against observations shown above (Fig. 3b and Table S3), then this difference hints a possible underestimation of BC column burden in the model over the western regions. However, it is difficult to draw a firm conclusion, given the likely differential role of dust in eastern vs western China. This differential likely also contributes to AAOD biases in modeling dust and may also impact biases in the satellite derived AI values."

Line 424-426: I think BC emissions from SC are also important for the column burdens over continental China in some seasons (e.g. JJA and SON), which needs to be outlined as well.

Response:

Thanks for the suggestion. We have now added the SC contribution to column burden, as "Column burdens of BC averaged over continental China mainly originate from emissions in North China, South China and outside China, with relative contributions ranging from 31–42%, 16–24% and 14–31%, respectively."

Line 443: "Figure S4a" should be replaced with "Fig. S5a".

Response:

Corrected.

Line 443-463: It is helpful for the authors to make supplemental plots showing the anomalous winds during polluted days that favor the accumulation of pollutants over each region.

Response:

   We did show in Fig. 8 the anomalous winds at 850 mb between polluted and normal days for each region during winter.

Line 505-509: Why the authors only choose the latitude range along longitude 150°E, not a domain covering East China Sea and West Pacific to quantitatively assess the BC contributions from China and outside China, similar to that impact over West United States?
Response:

   The outflow of aerosols, defined as the column-integrated aerosol flux or concentration along a vertical cross-section, is used to characterize the export of BC from East Asia. This calculation of outflow follows previous studies and thus is comparable to the results in these studies (Hadley et al., 2007; Matsui et al., 2013; Yang et al., 2015). In addition, using a region around 150°E does not change the values significantly (e.g., contribution from China changes from 53% for the outflow at150°E to 54% for an average over 145–155°E). We don't mean to assess the contributions from China and outside China to air quality over this region.

Line 509-510: I get lost here. It is not clear to me that 58% contribution from China emissions is for outflow or something else. Authors need to clarify this.
Response:

   Clarified as "The yearly contribution from emissions from China to outflow from East Asia in this study is 58%, similar to the contribution of 61% in Matsui et al. (2013) calculated based on eastward BC mass flux using WRF-CMAQ model with INTEX-B missions."

Line 531-538: I think the authors should list a table to compare your results with other studies, including annual BC emission budgets, burden, lifetime, DRF and DRF efficiency.
Response:

   Thanks the suggestion. We have added Table S5 to compare these values with previous studies.
   We have also added a discussion of this comparison in the manuscript, as "The total DRF of BC averaged over continental China simulated in this study is 2.27 W m$^{-2}$, larger than 0.64–1.55 W m$^{-2}$ in previous studies (Wu et al., 2008; Zhuang et al., 2011; Li et al., 2016), probably due to the different emissions in the time periods of study, as shown in Table S5." And "The annual mean and regional mean DRF efficiency in China is 0.91 W m$^{-2}$ Tg$^{-1}$, within the range of 0.41–1.55 W m$^{-2}$ Tg$^{-1}$ from the previous studies (Table S5)."

**Table S5.** Comparison of the simulated annual mean emission, burden, DRF and DRF efficiency in China in this study with the values reported in three previous studies.

| Reference | Model | Year | Emission in China (Gg yr$^{-1}$) | Burden (mg m$^{-2}$) | DRF (Wm$^{-2}$) | DRF efficiency (W m$^{-2}$ Tg$^{-1}$) |
|---|---|---|---|---|---|---|
| Wu et al. (2008) | RegCM3 | 2000 | 1005 | 0.55–1.42 | 0.64–1.55 | 0.64–1.55 |
| Zhuang et al. (2011) | RegCCMS | 2006 | 1811 | 1.12 | 0.75 | 0.41 |
| Li et al. (2016) | GEOS-Chem | 2010 | 1840 | | 1.22 | 0.66 |
| This study | CESM | 2010–2014 | 2497 | 1.45 | 2.27 | 0.91 |

Line 654-655: Other modeling studies also found model low bias over China using CAWNET, e.g. Huang et al., 2013; Wang et al., 2014, which can be referenced here.

Huang, Y., S. Wu, M. K. Dubey, and N. H. F. French, Impact of aging mechanism on model simulated carbonaceous aerosols, Atmos. Chem. Phys., 13, 6329–6343, doi:10.5194/acp-13- 6329-2013, 2013.

Wang, Q., D.J. Jacob, J.R Spackman, A.E. Perring, J.P. Schwarz, N. Moteki, E.A. Marais, C. Ge, J. Wang and S.R.H. Barrett, Global budget and radiative forcing of black carbon aerosol: constraints from pole-to-pole (HIPPO) observations across the Pacific, J. Geophys. Res., 119, 195- 206, 2014.

Response:
    Added.

Line 669: "and" is missing between "modeled" and "observed".
Response:
    Added.

References:

Zhang, L. M., Gong S. L., Padro J. and Barrie L.: A size-segregated particle dry deposition scheme for an atmospheric aerosol module, Atmos. Environ., 35, 549-560, doi:10.1016/S1352-2310(00)00326-5 ,2001.

Abdul-Razzak, H., and Ghan S. J.: A parameterization of aerosol activation: 2. Multiple aerosol types, J. Geophys. Res., 105, 6837–6844, doi:10.1029/1999JD901161, 2000.

Wang, H., R. C. Easter, P. J. Rasch, M. Wang, X. Liu, S. J. Ghan, Y. Qian, J.-H. Yoon, P.-L. Ma, and V. Vinoj (2013), Sensitivity of remote aerosol distributions to representation of cloud-aerosol interactions in a global climate model, Geosci. Model Dev., 6, 765–782, doi:10.5194/gmd-6-765-2013.

Hadley, O. L., V. Ramanathan, G. R. Carmichael, Y. Tang, C. E. Corrigan, G.

C. Roberts, and G. S. Mauger (2007), Trans-Pacific transport of black carbon and fine aerosols (D < 2.5 μm) into North America, J. Geophys. Res., 112, D05309, doi:10.1029/2006JD007632.

Matsui, H., M. Koike, Y. Kondo, N. Oshima, N. Moteki, Y. Kanaya, A. Takami, and M. Irwin (2013), Seasonal variations of Asian black carbon outflow to the Pacific: Contribution from anthropogenic sources in China and biomass burning sources in Siberia and Southeast Asia, J. Geophys. Res. Atmos., 118, 9948–9967, doi:10.1002/jgrd.50702.

Yang Y., H. Liao, and S. Lou (2015), Decadal trend and interannual variation of outflow of aerosols from East Asia: Roles of variations in meteorological parameters and emissions, Atmos. Environ., 100, 141-153, doi:10.1016/j.atmosenv.2014.11.004.

Oshima, N., Y. Kondo, Moteki N., Takegawa N., Koike M., Kita K., Matsui H., Kajino M., Nakamura H., Jung J. S., and Kim Y. J.: Wet removal of black carbon in Asian outflow: Aerosol Radiative Forcing in East Asia (A-FORCE) aircraft campaign, J. Geophys. Res., 117, D03204, doi:10.1029/2011JD016552, 2012.

Huang, Y., S. Wu, M. K. Dubey, and N. H. F. French, Impact of aging mechanism on model simulated carbonaceous aerosols, Atmos. Chem. Phys., 13, 6329–6343, doi:10.5194/acp-13- 6329-2013, 2013.

Wang, Q., D.J. Jacob, J.R Spackman, A.E. Perring, J.P. Schwarz, N. Moteki, E.A. Marais, C. Ge, J. Wang and S.R.H. Barrett, Global budget and radiative forcing of black carbon aerosol: constraints from pole-to-pole (HIPPO) observations across the Pacific, J. Geophys. Res., 119, 195- 206, 2014.

---

## Author Comment (AC3) · 2 Feb 2017

**Manuscript # acp-2016-1032**

**Responses to Reviewer #3**

This study quantified source contributions of black carbon (BC) mass concentrations, trans-Pacific transport of BC, and direct radiative forcing of BC from seven regions in China using the Community Earth System Model with a source-tagging technique. The authors showed that BC concentrations were dominated by local emissions for regions with high emissions (e.g., North China, South China), whereas non-local emissions were important for regions with low emissions (e.g., Northwest China, Tibetan Plateau). They also showed that emissions from China and other regions were equally important for the BC outflow from East Asia and that emissions from China would be important for air quality in western United States. The annual mean direct radiative forcing of BC in China in their simulations was 2.3 W m$^{-2}$, and the contribution from emissions in China was estimated to be 66%.

The purpose of this study is interesting and the results obtained in this study are important to understand BC behavior in the atmosphere over East Asia. I think the authors should describe several points (shown below) more clearly, but overall the manuscript is written well and is suitable for the publication of this journal.

Major comments:

(1) Importance of BC in air quality problems
The authors sometimes use BC as an indicator of pollution (or air quality) in China (Lines 39-40, Lines 101-102, Lines 429-431, and Lines 571-572). However, I think it is questionable whether the concentrations and/or source contributions of BC can be used to represent those of total aerosols. Inorganic and organic species are dominant in China during polluted days, and spatial/temporal variations and source contributions of these species are largely different from those of BC because spatial distributions of emissions (e.g., BC v.s. SO2) and formation processes (primary v.s. secondary) are considerably different. For example, Matsui et al. (2009) showed that primary aerosols around Beijing were controlled by emissions within 100 km around Beijing within the preceding 24 h, while emissions as far as 500 km and within the preceding 3 days were found to affect secondary aerosols. Therefore, it is not always correct to extend the results of BC (e.g., source contributions) to the discussions on pollution and air quality because inorganic and organic species could have larger contributions from non-local emissions than BC. Please consider this point and describe the limitation of using BC results only in the discussions of air quality problems. In addition, please show the

percentage of BC mass to total mass (PM2.5) in China in the manuscript.
Response:

We agree with the referee that by no means BC can represent the total $PM_{2.5}$ in air quality problems, which is not the focus of this study. We have added a paragraph in the discussion section to describe the limitation of using BC as a tracer for air pollution, as "In this study, BC is used as an indicator of pollution (or air quality) in China. Although BC is often co-emitted with other species, such as primary organic matter, organic gases and sulfuric gases, source-receptor relationship of BC may not fully represent that of total aerosols. The contribution of BC to total near-surface $PM_{2.5}$ concentrations averaged over China is less than 10%. Other aerosols, such as sulfate, are dominant in China during polluted days. The spatio-temporal variations and source contributions of these species are largely different from those of BC because spatial distributions of emissions (e.g., $SO_2$) and formation processes can be considerably different. For example, Matsui et al. (2009) showed that primary aerosols around Beijing were determined by emissions within 100 km around Beijing within the preceding 24 hours, while emissions as far as 500 km and within the preceding 3 days were found to affect secondary aerosols in Beijing. Thus, the secondary aerosols could have larger contributions from non-local emissions than BC. BC concentrations are highest in winter over China due to higher emissions, while sulfate concentrations reach maximum in summer when the strong sunlight and high temperature favor the sulfate formation. Therefore, knowing the accurate source attributions of air pollution in China requires source tagging for more aerosol species, such as sulfate."

(2) Treatment of optical property and CCN activity of BC (Lines 151-169)
I could not find the description on the treatment of optical property (well-mixed, core- shell, or others) and CCN activity (conversion from hydrophobic to hydrophilic BC) of BC in the MAM3 model. I assume that well-mixed optical treatment is used to calculate BC absorption and that all BC particles are treated as hydrophilic BC in MAM3. Please describe the treatment of optical property and CCN activity of BC in the manuscript, and add some description on the potential impact (uncertainty) of these treatments on the estimation of BC concentrations, trans-Pacific transport of BC, AAOD, and direct radiative forcing of BC and their source contributions.
Response:

Thanks for the suggestion. We have added more information regarding the treatment of BC in the model to the Methods section and also added a paragraph discussing the potential influence in the Methods and Conclusions and Discussions section, shown as below:

Aerosol optical properties for each mode are parameterized according to Ghan and Zaveri (2007). Refractive indices for aerosols are taken from the OPAC (optical properties for aerosols and clouds) software package (Koepke and Schult, 1998), but for BC at solar wavelengths the values are updated

from Bond and Bergstrom (2006). In MAM3, the aging process of BC is neglected by assuming the immediate mixing of BC with other aerosol species.

BC aging in the atmosphere is important for BC concentration and its optical properties, which transforms BC from hydrophobic aggregates to hydrophilic particles coated with soluble materials. He et al. (2015, 2016a) found that BC optical properties varied by a factor of two or more due to different coating structures during BC aging process based on their theoretical and experimental intercomparison. Oshima et al. (2009) and He et al. (2016b) pointed out that the use of various microphysical BC aging schemes could significantly improve simulations of BC concentrations compared to the simplified aging parameterizations. Liu et al. (2012) also reported that the wet removal rate of BC simulated in standard CAM5 is 60% higher than AeroCom multi-model mean due to the rapid or instantaneous aging of BC. H. Wang et al. (2013) showed that the explicit treatment of BC aging process with slow aging assumptions in CAM5 could significantly increase BC lifetime and the efficiency of BC long-range transport. In the three-mode aerosol module (MAM3) of CAM5 used in this study, the aging process of BC is neglected by assuming the immediate internal mixing of BC with other aerosol species in the same mode. This assumption could lead to an overestimation of wet removal of BC and, therefore, an underestimation of BC concentrations, absorption optical depth (Fig. 3) and direct radiative forcing. In addition, the internally-mixed optical treatment in CAM5 could also cause bias in BC absorption calculation. However, H. Wang et al. (2014) examined source-receptor relationships for BC under the different BC aging assumptions and found that the quantitative source attributions varied slightly while the qualitative source-receptor relationships still hold. Therefore, although the magnitude of simulated BC and its optical properties could be underestimated due to the instantaneous aging of BC and uncertainty in coating structures, we expect that the aging treatment in MAM3 of CAM5 should not influence the qualitative source attributions examined in this study.

Other comments:

(3) Line 70
Please describe the reason of the faster regional removal.
Response:
    Revised as "BC in East Asia has a shorter lifetime than the global mean value due to a faster regional removal (H. Wang et al., 2014), likely associated with a strong precipitation scavenging near sources and along the transport pathways over the Pacific Ocean."

(4) Lines 168-169
Please clarify the definition of the direct radiative forcing of BC. Is this

calculated from the difference of two radiative transfer calculations with and without BC for the clear-sky condition?

Response:

Revised as "Direct radiative forcing of BC is calculated as the difference in the top-of-the-atmosphere net radiative fluxes with and without BC for the all-sky condition following Ghan (2013)."

(5) Lines 182-204

Please show the difference of BC emission fluxes between the emission inventory used in this study and other emission inventories (e.g., INTEX-B, HTAP). The values are shown later (at Lines 534-538), but I think it is better to show them here. In addition, please add some comments on the impact of larger values of BC emissions in this study on the estimation of source contributions of BC. Can you add the values of BC emissions from outside China (e.g., India, Southeast Asia, Japan, Korea) to Figure 1b?

Response:

We have added Table S1 to compare the anthropogenic emissions used in this study with emissions from some previous studies. The anthropogenic emissions of BC in China in 2010–2014 are larger than those used in the previous studies for earlier years, partly as a result of the rapid increasing trend of BC in China during recent years. The higher emissions likely lead to higher concentrations and direct radiative forcing, and source contributions of BC in China, compared to the values reported in these studies. We have added these descriptions in the Methods section.

We also revised Fig. 1 to include BC emissions from outside China. Emissions at regional scale are summarized here instead of Country level because the model revolution is a bit course to characterize emissions by countries. Total BC emissions from neighboring regions including rest of East Asia (REA, with China excluded), South Asia (SAS), Southeast Asia (SEA), and Russia/Belarussia/Ukraine (RBU) are shown in Figure 1c. These source regions outside China are consistent with source regions defined in the second phase of Hemispheric Transport of Air Pollution (HTAP2). South Asia and Southeast Asia have relatively high emissions. They may dominate the contribution to concentrations and direct radiative forcing of BC in China, especially southern and western China, from foreign sources through long-range transport. We have also added these in the Methods section. We did not compare the emissions outside China with other studies, which is beyond the scope of this study. However, the comparison of CEDS emissions with other emission inventories can be found in Hoesly et al. (2017), which includes detailed information of the CEDS emissions and will be submit very soon.

We also added a paragraph about uncertainty in BC emissions in the Conclusions and discussions section, as "Uncertainty in China BC emissions has been estimated as –43% to 93% by Lu et al. (2011), –50% to 164% by Qin

and Xie (2012), ±176% by Kurokawa et al. (2013), and –28 to 126% by Zhao et al. (2013). The BC emissions estimates used here for China in 2010 are 40% higher than those of Zhao et al. (2013) and Lu et al. (2011) and 30% higher than Klimont et al. (2016), in large part due to a higher estimate of BC emissions from coal coke production. Emissions from coke production are particularly uncertain given that "there are no measurements for PM2.5 and BC emissions" (Huo et al. 2012) available to guide inventory estimates. Total rest of the world emissions other than China, which appear to be a major contributor to burdens over western regions, are within 1% of those from Klimont et al. (2016)."

**Table S1.** Comparison of CEDS annual mean anthropogenic BC emissions in China with those used in other studies

|  | Year | Anthropogenic emission in China (Gg/yr) |
|---|---|---|
| CEDS (Hoesly et al., 2016; this study) | 2010–2014 | 2467 |
| MIX (Li et al., 2017) | 2010 | 1765 |
| HTAP V2.2 (Janssens- Maenhout et al., 2015) | 2010 | 1741 |
| Lu et al. (2011) | 2010 | 1751 |
| Qin and Xie (2012) | 2009 | 1764 |
| Wang et al. (2012) | 2007 | 1879 |
| INTEX-B (Zhang et al., 2009) | 2006 | 1811 |

[Figure]

**Figure 1.** (a) Spatial distribution of annual mean total emissions (anthropogenic plus biomass burning, units: g C m$^{-2}$ yr$^{-1}$) of black carbon (BC) averaged over 2010–2014. The geographical BC source regions are selected as North China (NC, 109°E–east boundary, 30°–41°N), South China (SC, 109°E–east boundary, south boundary–30°N), Southwest China (SW, 100°–109°N, south boundary–32°N), Central-West China (CW, 100°–109°N, 32°N–north boundary), Northeast China (NE, 109°E–east boundary, 41°N–north boundary), Northwest China (NW, west boundary–100°E, 36°N–north boundary), and Tibetan Plateau (TP, west boundary–100°E, south boundary–36°N) in China and regions outside of China (RW, rest of the world). (b) Seasonal mean total emissions (units: Gg C, Gg = 10$^9$g) of BC from the seven BC source regions in China and (c) emissions from rest of East Asia (REA, with China excluded), South Asia (SAS), Southeast Asia (SEA), and Russia/Belarussia/Ukraine (RBU).

(6) Lines 261-263

You can show the contributions from outside China quantitatively from the tagged simulation results.

Response:

   We show the quantitative contributions from outside China in the Results section. It is not appropriate to directly compare the contribution to surface concentrations and that to column burden. Therefore, we decide to remove this sentence to avoid duplication and potential misunderstanding.

(7) Line 281

Please describe the reason of BC underestimation by up to a factor of 20.

Response:

   We have discussed possible causes of this underestimation of BC in the following part of the Results section and in the Discussion section as well:

   "Note that the model largely underestimates BC concentrations over China, compared to the observation, which has also been reported in many previous studies using different models and different emission inventories (e.g., Liu et al., 2012; Fu et al., 2012; Huang et al., 2013; H. Wang et al., 2013; Q. Wang et al., 2014; R. Wang et al., 2014; Li et al., 2016). One possible reason is that in situ measurements are point observations, while the model does not treat the subgrid variability of aerosols and assumes aerosols are uniformly distributed over the grid cell. R. Wang et al. (2014) found a reduction of negative bias (from −88% to −35%) in the modeled surface BC concentrations when using high-resolution emissions and modeling at 0.5° X 0.7° resolution. They find, however, that modeling over the North China Plain at an even higher resolution of 0.1°, further reduced the surface concentration bias there from 29% to 8%. This result indicates that the siting of observational stations can result in an artificial bias when comparing with relatively coarse model results. Further investigation of this siting/resolution bias is warranted, including investigation on whether this type of bias might extend, presumably to a lesser extent, also to AAOD measurements.

   Further reasons that could contribute to this bias are emission underestimation or inaccurate aerosol processes in the model. Given that the differences between modeled and observed AAOD over eastern China are relatively small (–18%), we conclude that, given current evidence, the total amount of atmospheric BC in these simulations is reasonable at least in this sub-region.

   Over eastern China, the BC concentrations are dominated by local emissions in this study, with local contribution of 64–93%. The underestimation of simulated BC concentrations over eastern China is more likely due to either underestimation of local emissions, too much aerosol removal within these regions, or resolution bias between observations and model grids. Over western China, 22–76% of the BC originates from emissions outside China.

Thus biases of simulated BC concentrations could also come from underestimation of emissions outside China and or too much removal of BC during long-range transport. Satellite data are a promising method to validate modeling and emissions inventories, given that they do not depend on the location of observing stations, providing more uniform spatial coverage. A comparison of modeled AAOD and satellite aerosol index (AI) provides an indication that the modeled burden in western China is underestimated, although the role of dust needs to be better characterized."

(8) Line 343
I cannot find large sources of BC in Northwest China in Figure 1a. Does the description here mean that there may be large sources of BC which are not considered in the emission inventory? Can you show the contribution of BC and dust to AAOD (in model) over this region? I think dust is dominant over this region.
Response:
    Yes, underestimation of emissions over Northwest China could be part of the story. In the source attribution analysis, we found that emissions from outside China also contribute substantially to BC concentrations in Northwest China. Therefore the underestimation in BC concentrations could also come from low biases in emissions outside China or too much removal during along the transport pathways in the model.
    Dust AAOD is indeed dominant over Northwest China (Fig. A). Potential biases in dust simulation could also lead to the difference in AAOD between model and observation. That is why we noted the potential bias from dust simulation, as "However, it is difficult to draw a firm conclusion, given the likely differential role of dust that also contributes to AAOD and AI, biases in modeling dust, and possible biases in the satellite derived AI values." And "A comparison of modeled AAOD and satellite aerosol index (AI) provides an indication that the modeled burden in western China is underestimated, although the role of dust needs to be better characterized."

[Figure]

Figure A. Simulated annual mean AAOD of BC and dust.

(9) Lines 667-669

Related to the comment (2), is an internally-mixed treatment used in the calculations of AAOD? If so, AAOD should be lower (underestimated more) when more realistic BC mixing state treatment is used in the optical calculations.

Response:

CAM5 assumes internal mixing between BC and other components in the same mode, but external mixing between modes in the calculations of optical properties of the mixture of aerosol particles. Compared to the more sophisticated shell-core treatment, the current treatment might underestimate the absorption of aerosol. The aerosol mixing state along with the representation of size distribution is still quite uncertain. There is no strong evidence suggesting which one is more realistic, so we prefer not to draw such a conclusion here.

References:

Matsui, H., et al. (2009), Spatial and temporal variations of aerosols around Beijing in summer 2006: Model evaluation and source apportionment, J. Geophys. Res., 114, D00G13, doi:10.1029/2008JD010906.

Ghan, S. J., and R. A. Zaveri (2007), Parameterization of optical properties for hydrated internally mixed aerosol, J. Geophys. Res., 112, D10201, doi:10.1029/2006JD007927.

Bond, T. C., and R. W. Bergstrom, Light absorption by carbonaceous particles: An investigative review, Aerosol. Sci. Technol., 40, 27–67, doi:10.1080/02786820500421521, 2006.

Koepke, M. H. P., and I. Schult, Optical properties of aerosols and clouds: The software package opac, Bull. Am. Meteorol. Soc., 79, 831–844, 1998, doi:10.1175/1520-0477(1998)079<0831:OPOAAC>2.0.CO;2.

He, C., Liou, K.-N., Takano, Y., Zhang, R., Levy Zamora, M., Yang, P., Li, Q., and Leung, L. R.: Variation of the radiative properties during black carbon aging: theoretical and experimental intercomparison, Atmos. Chem. Phys., 15, 11967-11980, doi:10.5194/acp-15-11967-2015, 2015.

He, C., Takano, Y., Liou, K.-N., Yang, P., Li, Q., and Mackowski, D. W.: Intercomparison of the GOS approach, superposition T- matrix method, and laboratory measurements for black carbon optical properties during aging, J. Quant. Spectrosc. Ra., 184, 287–296, doi:10.1016/j.jqsrt.2016.08.004, 2016a.

He, C., Li, Q., Liou, K.-N., Qi, L., Tao, S., and Schwarz, J. P.: Microphysics-based black carbon aging in a global CTM: constraints from HIPPO observations and implications for global black carbon budget, Atmos. Chem. Phys., 16, 3077-3098, doi:10.5194/acp-16-3077-2016, 2016b.

Oshima, N., M. Koike, Y. Zhang, Y. Kondo, N. Moteki, N. Takegawa, and Y. Miyazaki (2009), Aging of black carbon in outflow from anthropogenic sources using a mixing state resolved model: Model development and evaluation, J. Geophys. Res., 114, D06210, doi:10.1029/2008JD010680.

Liu, X., et al. (2012), Toward a minimal representation of aerosols in climate models: Description and evaluation in the Community Atmosphere Model CAM5, Geosci. Model Dev., 5, 709–739, doi:10.5194/gmd-5-709-2012.

Liu, X., Ma, P.-L., Wang, H., Tilmes, S., Singh, B., Easter, R. C., Ghan, S. J., and Rasch, P. J.: Description and evaluation of a new four-mode version of the Modal Aerosol Module (MAM4) within version 5.3 of the Community Atmosphere Model, Geosci. Model Dev., 9, 505-522, doi:10.5194/gmd-9-505-2016, 2016.

Wang, H., R. C. Easter, P. J. Rasch, M. Wang, X. Liu, S. J. Ghan, Y. Qian, J.-H. Yoon, P.-L. Ma, and V. Vinoj (2013), Sensitivity of remote aerosol distributions to representation of cloud-aerosol interactions in a global climate model, Geosci. Model Dev., 6, 765–782,

doi:10.5194/gmd-6-765-2013.

Wang, H., P. J. Rasch, R. C. Easter, B. Singh, R. Zhang, P.-L. Ma, Y. Qian, S. J. Ghan, and N. Beagley (2014), Using an explicit emission tagging method in global modeling of source-receptor relationships for black carbon in the Arctic: Variations, sources, and transport pathways, J. Geophys. Res. Atmos., 119, 12,888–12,909, doi:10.1002/ 2014JD022297.

Ghan, S. J. (2013), Technical Note: Estimating aerosol effects on cloud radiative forcing, Atmos. Chem. Phys., 13, 9971-9974, doi:10.5194/acp-13-9971-2013.

Li, K., Liao, H., Mao, Y. H., and Ridley, D. A.: Source sector and region contributions to concentration and direct radiative forcing of black carbon in China, Atmos. Environ., 124, 351–366, doi:10.1016/j.atmosenv.2015.06.014, 2016.

Janssens-Maenhout, G., Crippa, M., Guizzardi, D., Dentener, F., Muntean, M., Pouliot, G., Keating, T., Zhang, Q., Kurokawa, J., Wankmüller, R., Denier van der Gon, H., Kuenen, J. J. P., Klimont, Z., Frost, G., Darras, S., Koffi, B., and Li, M.: HTAP_v2.2: a mosaic of regional and global emission grid maps for 2008 and 2010 to study hemispheric transport of air pollution, Atmos. Chem. Phys., 15, 11411-11432, doi:10.5194/acp-15-11411-2015, 2015.

Lu, Z., Zhang, Q., and Streets, D. G.: Sulfur dioxide and primary carbonaceous aerosol emissions in China and India, 1996–2010, Atmos. Chem. Phys., 11, 9839-9864, doi:10.5194/acp-11-9839-2011, 2011.

Qin, Y. and Xie, S. D.: Spatial and temporal variation of anthropogenic black carbon emissions in China for the period 1980–2009, Atmos. Chem. Phys., 12, 4825-4841, doi:10.5194/acp-12-4825-2012, 2012.

Wang, R., Tao, S., Wang, W., Liu, J., Shen, H., Shen, G., Wang, B., Liu, X., Li, W., Huang, Y., Zhang, Y., Lu, Y., Chen, H., Chen, Y., Wang, C., Zhu, D., Wang, X., Li, B., Liu, W., Ma, J.: Black carbon emissions in China from 1949 to 2050, Environ. Sci. Technol., 46, 7595-7603, doi:10.1021/es3003684, 2012.

Zhang, Q., Streets, D. G., Carmichael, G. R., He, K. B., Huo, H., Kannari, A., Klimont, Z., Park, I. S., Reddy, S., Fu, J. S., Chen, D., Duan, L., Lei, Y., Wang, L. T., and Yao, Z. L.: Asian emissions in 2006 for the NASA INTEX-B mission, Atmos. Chem. Phys., 9, 5131-5153, doi:10.5194/acp-9-5131-2009, 2009.

---

## Author Response (AR1)

**Manuscript # acp-2016-1032**

**Responses to Reviewer #1**

The authors used the CESM model with a source-tagging method to quantify the source attributions for BC direct radiative forcing (DRF) and concentration as well as polluted events. They found that in addition to regional emissions within China, emissions outside China also contribute to a large portion of BC DRF over China. This study could improve the understanding on BC pollution in China and provide implications for policy makers. Before this manuscript can be considered for publication, I have a few comments that need to be addressed by the authors.

1. One critical factor influencing BC direct radiative effect is its optical properties (absorption and extinction cross section, asymmetry factor, and single scattering albedo). Recent studies (e.g., He et al. 2015, 2016b) showed that BC optical properties vary significantly (by up to more than a factor of two) due to different coating structures and aging stages during BC aging process, which further affects direct radiative effect. (1) Could the authors add some discussions on this aspect with reference to these recent studies, for example, potential uncertainty in their results caused by this factor? (2) Could this variation in BC optical properties due to coating structures contribute to the model biases in BC AAOD simulations as discussed in the first paragraph of Section 3.2? (3) It would be helpful if the authors could add more details on how the MAM3 model computes BC optical properties. For example, does it assume a core-shell structure for internally mixed BC?
References:
He, C., et al. : Variation of the radiative properties during black carbon aging: theoretical and experimental intercomparison, Atmos. Chem. Phys., 15, 11967-11980, doi:10.5194/acp-15-11967-2015, 2015.
He, C., et al. : Intercomparison of the GOS approach, superposition T-matrix method, and laboratory measurements for black carbon optical properties during aging, J. Quant. Spectrosc. Radiat. Transf., 184, 287–296, doi:10.1016/j.jqsrt.2016.08.004, 2016b.

Response:
    (1) Thanks for the suggestion. We have added discussions of the influence of aging processes on BC optical properties to the Conclusions and Discussions section, along with the references provided by the referee, as "BC aging in the atmosphere is important for BC concentration and its optical properties, which transforms BC from hydrophobic aggregates to hydrophilic particles coated with soluble materials. He et al. (2015, 2016a) found that BC optical properties varied by a factor of two or more due to different coating structures during BC aging process based on their theoretical and experimental intercomparison. Oshima et al. (2009) and He et al. (2016b) pointed out that the use of various microphysical BC aging schemes could significantly improve simulations of BC concentrations compared to the simplified aging parameterizations. Liu et al. (2012) also reported that the wet
removal rate of BC simulated in standard CAM5 is 60% higher than AeroCom
multi-model mean due to the rapid or instantaneous aging of BC. H. Wang et al.
(2013) showed that the explicit treatment of BC aging process with slow aging
assumptions in CAM5 could significantly increase BC lifetime and the efficiency of BC
long-range transport. In the three-mode aerosol module (MAM3) of CAM5 used in this
study, the aging process of BC is neglected by assuming the immediate internal
mixing of BC with other aerosol species in the same mode. This assumption could
lead to an overestimation of wet removal of BC and, therefore, an underestimation of
BC concentrations, absorption optical depth (Fig. 3) and direct radiative forcing. In
addition, the internally-mixed optical treatment in CAM5 could also cause bias in BC
absorption calculation. However, H. Wang et al. (2014) examined source-receptor
relationships for BC under the different BC aging assumptions and found that the
quantitative source attributions varied slightly while the qualitative source-receptor
relationships still hold. Therefore, although the magnitude of simulated BC and its
optical properties could be underestimated due to the instantaneous aging of BC and
uncertainty in coating structures, we expect that the aging treatment in MAM3 of
CAM5 should not influence the qualitative source attributions examined in this study."
(2) We agree that uncertainties in BC optical properties due to coating structures
and/or aging could contribute to model biases in simulated BC concentrations and
AAOD. We have added this statement in the discussion section. Please see the
revisions above.
(3) In the MAM3 aerosol module of CAM5 used in this study, the aging process of
BC is neglected by assuming an instantaneous mixing of BC with other aerosol
species in the same accumulation mode, which has been added in the discussion
section. We also add a more detailed description of the calculation of aerosol optical
properties in the Methods section, as "Aerosol optical properties for each mode are
parameterized according to Ghan and Zaveri (2007). Refractive indices for aerosols
are taken from the OPAC (optical properties for aerosols and clouds) software
package (Koepke and Schult, 1998), but for BC at solar wavelengths the values are
updated from Bond and Bergstrom (2006)."
2. For BC emissions, a number of global and regional emission inventories have been
developed, which showed large uncertainties and differences among each other
(e.g., Fig. 4 in Wang et al., 2014). It would be helpful if the authors could discuss the
uncertainty associated with the emission inventory used in this study and how this
inventory compares with previous ones for both inside and outside China, since the
authors pointed out that emissions outside China also contribute a lot to BC DRF in
China.
Reference:
Wang, R., et al. : Trend in Global Black Carbon Emissions from 1960 to 2007,
Environ. Sci. Technol., 48, 6780−6787, doi: 10.1021/es5021422, 2014.

Response:
Uncertainty in China BC emissions has been estimated as –43% to 93% by Lu et
al. (2011), –50% to 164% by Qin and Xie (2012), ±176% by Kurokawa et al. (2013),
and –28 to 126% by Zhao et al. (2013). The BC emissions estimates used here for
China in 2010 are 40% higher than those of Zhao et al. (2013) and Lu et al. (2011)
and 30% higher than Klimont et al. (2016), in large part due to a higher estimate of
BC emissions from coal coke production. Emissions from coke production are
particularly uncertain given that "there are no measurements for $PM_{2.5}$ and BC
emissions" (Huo et al. 2012) available to guide inventory estimates. Total rest of the
world emissions other than China, which appear to be a major contributor to burdens
over western regions, are within 1% of those from Klimont et al. (2016). We have
added these discussions in Conclusions and discussions section.
We have added Table S1 to compare the anthropogenic emissions used in this
study with emissions from some previous studies. The anthropogenic emissions of
BC in China in 2010–2014 are larger than those used in the previous studies for
earlier years, partly as a result of a higher estimate of BC emissions from coal coking
production. The higher emissions likely lead to higher concentrations and direct
radiative forcing, and source contributions of BC in China, compared to the values
reported in these studies. We have added these descriptions in the Methods section.
We also revised Fig. 1 to include BC emissions from outside China. Emissions at
regional scale are summarized here instead of Country level because the model
revolution is a bit course to characterize emissions by countries. Total BC emissions
from neighboring regions including rest of East Asia (REA, with China excluded),
South Asia (SAS), Southeast Asia (SEA), and Russia/Belarussia/Ukraine (RBU) are
shown in Figure 1c. These source regions outside China are consistent with source
regions defined in the second phase of Hemispheric Transport of Air Pollution
(HTAP2). South Asia and Southeast Asia have relatively high emissions. They may
dominate the contribution to concentrations and direct radiative forcing of BC in
China, especially southern and western China, from foreign sources through
long-range transport. We have also added these in the Methods section. We did not
compare the emissions outside China with other studies, which is beyond the scope
of this study. However, the comparison of CEDS emissions with other emission
inventories can be found in Hoesly et al. (2017), which includes detailed information
of the CEDS emissions and will be submit very soon.

**Table S1.** Comparison of CEDS annual mean anthropogenic BC emissions in China
with those used in other studies

|  | Year | Anthropogenic emission in China (Gg/yr) |
|---|---|---|
| CEDS (Hoesly et al., 2016; this study) | 2010–2014 | 2467 |
| MIX (Li et al., 2017) | 2010 | 1765 |
| HTAP V2.2 (Janssens-Maenhout et al., 2015) | 2010 | 1741 |
| Lu et al. (2011) | 2010 | 1751 |
| Qin and Xie (2012) | 2009 | 1764 |
| Wang et al. (2012) | 2007 | 1879 |
| INTEX-B (Zhang et al., 2009) | 2006 | 1811 |

[Figure]

[Figure]

**Figure 1.** (a) Spatial distribution of annual mean total emissions (anthropogenic plus biomass burning, units: g C m$^{-2}$ yr$^{-1}$) of black carbon (BC) averaged over 2010–2014. The geographical BC source regions are selected as North China (NC, 109°E–east boundary, 30°–41°N), South China (SC, 109°E–east boundary, south boundary–30°N), Southwest China (SW, 100°–109°N, south boundary–32°N), Central-West China (CW, 100°–109°N, 32°N–north boundary), Northeast China (NE, 109°E–east boundary, 41°N–north boundary), Northwest China (NW, west boundary–100°E, 36°N–north boundary), and Tibetan Plateau (TP, west boundary–100°E, south boundary–36°N) in China and regions outside of China (RW, rest of the world). (b) Seasonal mean total emissions (units: Gg C, Gg = 10$^9$g) of BC from the seven BC source regions in China and (c) emissions from rest of East Asia (REA, with China excluded), South Asia (SAS), Southeast Asia (SEA), and Russia/Belarussia/Ukraine (RBU).

3. Another important factor affecting BC simulations is aging process, which directly alters BC wet scavenging and lifetime. As pointed out by some recent studies (e.g., Oshima et al., 2009; He et al., 2016a), applying microphysical BC aging schemes could significantly improve simulations of BC concentrations compared with simplified aging parameterizations. (1) Could the authors briefly describe how BC aging is treated/computed in their model? (2) It would be helpful if the authors could briefly discuss the BC aging effect on their results with reference to these recent studies. (3) The authors mentioned in Lines 255–256 that model biases in BC concentration over China is likely due to inaccurate emissions and wet scavenging. Could this bias also be caused by model uncertainty related to BC aging? Some discussions would be useful.

reference:

He, C., Li, Q., Liou, K.-N., Qi, L., Tao, S., and Schwarz, J. P.: Microphysics-based black carbon aging in a global CTM: constraints from HIPPO observations and implications for global black carbon budget, Atmos. Chem. Phys., 16, 3077-3098, doi:10.5194/acp-16-3077-2016, 2016a.

Oshima, N., et al. : Aging of black carbon in outflow from anthropogenic sources using a mixing state resolved model: Model development and evaluation, J. Geophys. Res., 114, D06210, doi:10.1029/2008JD010680, 2009.

Response:
    (1) Thanks for the suggestion. We had now added more description of the BC aging treatment and related discussions to the paper. Please see the response to comment #1 above.

(2) Following the referee's suggestion, we had now added more discussions in this regard. Please see the response to comment #1 above.

(3) Yes, the overestimation of wet scavenging is partly caused by the assumption of instantaneous aging of BC in the model. We have added the message to the manuscript, as "Larger wet removal rate and shorter lifetime of aerosols along with the instantaneous aging of BC in the MAM3 can also lead to the lower concentrations of BC (e.g., Wang et al., 2011; Liu et al., 2012; H. Wang et al., 2013; Kristiansen et al., 2016)." and in discussion section as "This assumption could lead to an overestimation of wet removal of BC and, therefore, an underestimation of BC concentrations, absorption optical depth (Fig. 3) and direct radiative forcing."

4. The authors derived BC AAOD from AERONET observations by using the method in Bond et al. (2013). However, a recent study by Schuster et al. (2016) pointed out some weaknesses and problems related to the Bond et al. (2013) method. Could the author briefly discuss this issue? How would this affect the results in this study?

Reference:
Schuster, G. L., et al. : Remote sensing of soot carbon – Part 2: Understanding the absorption Ångström exponent, Atmos. Chem. Phys., 16, 1587–1602, doi:10.5194/acp-16-1587-2016, 2016.

Response:
We have included a discussion of this caveat associated with the BC AAOD comparison, as "Note that, the observed AAOD of BC is derived from AERONET measurements using the absorption Ångström exponent. A recent study (Schuster et al., 2016) reported that absorption Ångström exponent is not a robust parameter for separating out carbonaceous absorption in the AERONET database, which could cause biases in the AAOD estimates."

References:

[revised manuscript text omitted]

**Responses to Reviewer #2**

Yang et al. investigated the BC source attributions (or more specifically, region attributions) in China with a source-tagging technique by employing a global climate model, NCAR's Community Earth System Model. They found out that BC emissions from local (inside China) and non-local (outside China) are both generally important contributions to air quality in different regions of China, BC outflow from East Asia and direct radiative forcings. Overall, this paper is a helpful addition to our community that attempts to improve our understanding of BC source-receptor relationship. This paper generally reads well and is within the scope of ACP. However, before it can be accepted for publication in ACP, I have several comments that need to be properly addressed.

Major comments:

An important part of this study was to quantify the BC source contributions to trans-boundary and trans-pacific transport. In terms of model performance evaluation, this study only validated model simulations with observed BC surface concentrations from CAWNET and AAOD from AERONET over China. We don't know the efficiency of BC outflow from East Asia. In this paper, it obviously missed the evaluations of model simulated vertical profiles of BC against aircraft campaign observations, e.g. A-FORCE and HIPPO, which should be employed to compare with model simulations.

Response:
Thanks for the suggestion. The simulated BC vertical profile in CAM5 has been extensively evaluated in many previous studies. Liu et al. (2012) compared the observed and simulated BC vertical profiles in the tropics, middle latitudes, and high latitudes from six aircraft campaigns: AVE Houston (NASA Houston Aura Validation Experiment), CR-AVE (NASA Costa Rica Aura Validation Experiment), TC4 (Tropical Composition, Cloud and Climate Coupling), CARB (NASA initiative in collaboration with California Air Resources Board), ARCTAS (NASA Arctic Research of the Composition of the Troposphere from Aircraft and Satellite), and ARCPAC (NOAA Aerosol, Radiation, and Cloud Processes affecting Arctic Climate), as well as BC vertical profiles over the Arctic Ocean and the remote Pacific Ocean during the HIPPO (HIAPER Pole-to-Pole Observations) campaign. They found that measured BC mixing ratios showed a strong gradient from the boundary layer to the free troposphere in the tropics, while modeled BC mixing ratios showed a smaller decrease with altitude in the free troposphere, thus overestimating observations above 500 hPa. Compared to HIPPO campaign, the CAM5 model captured the vertical variations of BC mixing ratio reasonably well in the SH high latitudes and NH

and SH mid-latitudes. However, modeled BC showed less vertical reduction in the
tropics, thus significantly overestimating BC in the upper troposphere.
Wang et al. (2013) implemented in CAM5 a unified treatment of wet removal and
vertical transport of aerosols by convection, which included an explicit secondary
activation of aerosols being laterally entrained into convective clouds above cloud
base. The comparisons between the new CAM5 simulated vertical profiles of BC
mass mixing ratios and the HIPPO and the field campaign aircraft observations
showed a substantial improvement in the simulation of BC in mid- and upper
troposphere, where the excessive BC was significantly reduced. All of these key
model improvements by Wang et al. (2013) have been included in the version of
CAM5 being used in the present study. Therefore, we did not duplicate the evaluation
of BC vertical profiles with HIPPO observation. We have now revised the description
before model evaluation to make it clear, as "The simulations of aerosols, especially
BC, using CAM5 have been extensively evaluated against observations including
aerosol mass and number concentrations, vertical profiles, aerosol optical properties,
aerosol deposition, and cloud-nucleating properties in several previous studies (e.g.,
Liu et al., 2012, 2016; H. Wang et al., 2013; Ma et al., 2013b; Jiao et al., 2014; Qian
et al., 2014; R. Zhang et al., 2015a,b)."
In addition, as the referee suggested, we have added a comparison of the
simulated BC vertical profile with A-FORCE measurements over East Asia (see
Figure S2). The model successfully reproduces the vertical profile of BC. The bias is
relatively small. We have also added a relevant discussion to the revised manuscript,
as "Figure S2 compares the observed and simulated vertical profiles of BC
concentrations in the East-Asian outflow region. The model successfully reproduces
the vertical profile of BC that was measured in March–April 2009 during the
A-FORCE field campaign and reported by Oshima et al. (2012)."

[Figure]

Figure S2. Observed and simulated mean vertical profiles of BC concentrations in the
East-Asian outflow region. The observed BC profile is from the A-FORCE field campaign conducted over the Yellow Sea, the East China Sea, and the western
Pacific Ocean in March–April 2009 (Oshima et al., 2012).

Other minor comments:

Line 124: the reference Hoesly et al., 2016 is missing in the reference list.
Response:
   The paper is still in preparation. A draft can be made available to referees upon
request.

Line 162: A brief description of dry/wet deposition scheme for BC in CAM is lacking
here, especially the wet scavenging and how it is improved following H. Wang et al.
(2013).
Response:
   Aerosol dry deposition velocities are calculated using the Zhang et al. (2001)
parameterization. The wet deposition of aerosols in our CAM5 model includes
in-cloud wet removal (i.e., activation of interstitial aerosols to cloud-borne particles
followed by precipitation scavenging) and below-cloud wet removal (i.e., capture of
interstitial aerosol particles by falling precipitation particles) for both stratiform and
convective clouds. Aerosol activation is calculated with the parameterization of
Abdul-Razzak and Ghan (2000) for stratiform cloud throughout the column and
convective cloud at cloud base, while the secondary activation above convective
cloud base has a simpler treatment with an assumed maximum supersaturation in
convective updrafts (H. Wang et al., 2013). The unified treatment for convective
transport and aerosol wet removal along with the explicit aerosol activation above
convective cloud base was developed by H. Wang et al. (2013) and included in the
CAM5 version being used in this study. As discussed in the response to the major
comment, this implementation reduces the excessive BC aloft and better simulates
observed BC concentrations in the mid- to upper-troposphere.
   We have now added these descriptions to the Methods section.

Line 334-336: This sentence should be corrected as "AI derived from Total Ozone
Mapping Spectrometer (TOMS) measurements also shows similar pattern as
simulate AAOD (Fig. S2)."
Response:
   Corrected.

Line 339-344: What's the assumption here? Why the ratio of AAOD to AI should be
the same between western and eastern China? What is the role of dust here in
assisting the speculation?
Response:

Based on the comparison of AAOD between the model simulation and
AERONET, we found that the model reproduced well the observed AAOD over
eastern China. Therefore, we assume that the ratio of modeled AAOD and satellite AI
(indicator for absorbing aerosols) is correct over eastern China. The AAOD/AI over
eastern China is much larger than western China, suggesting that the ratio AAOD/AI
is lower than the true value and AAOD or BC burden is likely underestimated in the
model. Both AAOD and AI represent absorbing aerosols, the ratio of AAOD to AI may
be different but similar between different regions in China. The difference in the ratio
between eastern and western China is quite large, suggesting the existence of a
significant bias. This is consistent with the contrast of biases in near-surface BC
concentrations between the two regions (shown in Fig. 3). However, both BC and
dust can contribute to AAOD and AI. Potential biases in the modeled dust could also
lead to the inconsistence of AAOD/AI between eastern and western China.
We have revised the description to make it clear, as "If we assume that the
simulated AAOD do not have large biases over eastern China based on the
evaluation against observations shown above (Fig. 3b and Table S3), then this
difference hints a possible underestimation of BC column burden in the model over
the western regions. However, it is difficult to draw a firm conclusion, given the likely
differential role of dust in eastern vs western China. This differential likely also
contributes to AAOD biases in modeling dust and may also impact biases in the
satellite derived AI values."
Line 424-426: I think BC emissions from SC are also important for the column
burdens over continental China in some seasons (e.g. JJA and SON), which needs to
be outlined as well.
Response:
Thanks for the suggestion. We have now added the SC contribution to column
burden, as "Column burdens of BC averaged over continental China mainly originate
from emissions in North China, South China and outside China, with relative
contributions ranging from 31–42%, 16–24% and 14–31%, respectively."
Line 443: "Figure S4a" should be replaced with "Fig. S5a".
Response:
Corrected.
Line 443-463: It is helpful for the authors to make supplemental plots showing the
anomalous winds during polluted days that favor the accumulation of pollutants over
each region.
Response:
We did show in Fig. 8 the anomalous winds at 850 mb between polluted and
normal days for each region during winter.
Line 505-509: Why the authors only choose the latitude range along longitude 150°E,
not a domain covering East China Sea and West Pacific to quantitatively assess the

BC contributions from China and outside China, similar to that impact over West
United States?
Response:
The outflow of aerosols, defined as the column-integrated aerosol flux or
concentration along a vertical cross-section, is used to characterize the export of BC
from East Asia. This calculation of outflow follows previous studies and thus is
comparable to the results in these studies (Hadley et al., 2007; Matsui et al., 2013;
Yang et al., 2015). In addition, using a region around 150°E does not change the
values significantly (e.g., contribution from China changes from 53% for the outflow
at150°E to 54% for an average over 145–155°E). We don't mean to assess the
contributions from China and outside China to air quality over this region.
Line 509-510: I get lost here. It is not clear to me that 58% contribution from China
emissions is for outflow or something else. Authors need to clarify this.
Response:
Clarified as "The yearly contribution from emissions from China to outflow from
East Asia in this study is 58%, similar to the contribution of 61% in Matsui et al. (2013)
calculated based on eastward BC mass flux using WRF-CMAQ model with INTEX-B
missions."
Line 531-538: I think the authors should list a table to compare your results with other
studies, including annual BC emission budgets, burden, lifetime, DRF and DRF
efficiency.
Response:
Thanks the suggestion. We have added Table S5 to compare these values with
previous studies.
We have also added a discussion of this comparison in the manuscript, as "The
total DRF of BC averaged over continental China simulated in this study is 2.27 W
$m^{-2}$, larger than 0.64–1.55 W $m^{-2}$ in previous studies (Wu et al., 2008; Zhuang et al.,
2011; Li et al., 2016), probably due to the different emissions in the time periods of
study, as shown in Table S5." And "The annual mean and regional mean DRF
efficiency in China is 0.91 W $m^{-2}$ $Tg^{-1}$, within the range of 0.41–1.55 W $m^{-2}$ $Tg^{-1}$ from
the previous studies (Table S5)."
**Table S5.** Comparison of the simulated annual mean emission, burden, DRF and
DRF efficiency in China in this study with the values reported in three previous
studies.

| Reference | Model | Year | Emission in China (Gg $yr^{-1}$) | Burden (mg $m^{-2}$) | DRF (W$m^{-2}$) | DRF efficiency (W $m^{-2}$ $Tg^{-1}$) |
|---|---|---|---|---|---|---|
| Wu et al. (2008) | RegCM3 | 2000 | 1005 | 0.55–1.42 | 0.64–1.55 | 0.64–1.55 |
| Zhuang et al. (2011) | RegCCMS | 2006 | 1811 | 1.12 | 0.75 | 0.41 |
| Li et al. (2016) | GEOS-Chem | 2010 | 1840 | | 1.22 | 0.66 |
| This study | CESM | 2010–2014 | 2497 | 1.45 | 2.27 | 0.91 |

Line 654-655: Other modeling studies also found model low bias over China using CAWNET, e.g. Huang et al., 2013; Wang et al., 2014, which can be referenced here. Huang, Y., S. Wu, M. K. Dubey, and N. H. F. French, Impact of aging mechanism on model simulated carbonaceous aerosols, Atmos. Chem. Phys., 13, 6329–6343, doi:10.5194/acp-13- 6329-2013, 2013.
Wang, Q., D.J. Jacob, J.R Spackman, A.E. Perring, J.P. Schwarz, N. Moteki, E.A. Marais, C. Ge, J. Wang and S.R.H. Barrett, Global budget and radiative forcing of black carbon aerosol: constraints from pole-to-pole (HIPPO) observations across the Pacific, J. Geophys. Res., 119, 195- 206, 2014.
Response:
    Added.

Line 669: "and" is missing between "modeled" and "observed".
Response:
    Added.

Response:

Thanks for the suggestion. We have added a paragraph in the discussion section
to describe the limitation of using BC results, as "In this study, BC is used as an
indicator of pollution (or air quality) in China. It should be noted that, contributions of
BC from different source regions may not fully represent source-receptor relationship
of total aerosols. The contribution of BC to total near-surface PM2.5 (sum of BC,
sulfate, primary organic matter and second organic carbon) concentration averaged
over China is only about 10%. Inorganic and organic species aerosols, such as
sulfate, are dominant in China during polluted days, and spatial/temporal variations
and source contributions of these species are largely different from those of BC
because spatial distributions of emissions (e.g., BC v.s. $SO_2$) and formation
processes (primary v.s. secondary) are considerably different. For example, Matsui et
al. (2009) showed that primary aerosols around Beijing were controlled by emissions
within 100 km around Beijing within the preceding 24 h, while emissions as far as 500
km and within the preceding 3 days were found to affect secondary aerosols. Thus,
the inorganic and organic species could have larger contributions from non-local
emissions than BC. BC concentrations are highest in winter over China due to higher
emissions, while sulfate concentrations reach maximum in summer because of
stronger sunlight and higher temperature preferring sulfate formation. Therefore,
knowing the accurate source attribution of air pollution in China requires source
tagging for more aerosol species, such as sulfate."
(2) Treatment of optical property and CCN activity of BC (Lines 151-169)
I could not find the description on the treatment of optical property (well-mixed, core-
shell, or others) and CCN activity (conversion from hydrophobic to hydrophilic BC) of
BC in the MAM3 model. I assume that well-mixed optical treatment is used to
calculate BC absorption and that all BC particles are treated as hydrophilic BC in
MAM3. Please describe the treatment of optical property and CCN activity of BC in
the manuscript, and add some description on the potential impact (uncertainty) of
these treatments on the estimation of BC concentrations, trans-Pacific transport of
BC, AAOD, and direct radiative forcing of BC and their source contributions.
Response:
Thanks for the suggestion. We have added all these information in methods
section and added a paragraph discussing the potential influence in conclusions and
discussions section shown as below:
Aerosol optical properties for each mode are internally-mixed and parameterized
according to Ghan and Zaveri (2007). Refractive indices for aerosols are taken from
OPAC (Koepke and Schult, 1998), but for solar wavelengths of BC the value is from
Bond and Bergstrom (2006). In MAM3, the aging process of BC is neglected by
assuming the immediate mixing of BC with other aerosol species.
BC aging in the atmosphere is important for BC concentration and its optical
properties, which transforms BC from hydrophobic aggregates to hydrophilic particles
coated with soluble materials. He et al. (2015, 2016a) found BC optical properties
varied by up to more than a factor of two due to different coating structures and aging
stages during BC aging process based on theoretical and experimental intercomparison. Oshima et al. (2009) and He et al. (2016b) pointed out that applying
microphysical BC aging schemes could significantly improve simulations of BC
concentrations compared with simplified aging parameterizations. Liu et al. (2012)
also reported that the wet removal rate of BC simulated in standard CAM5 is 60%
higher than AeroCom multi-model mean due to rapid or instantaneous aging of BC.
H. Wang et al. (2013) showed that the explicit treatment of BC aging process with
slower aging assumptions in CAM5 could significantly increase BC lifetime and the
efficiency of BC long-range transport. In MAM3 aerosol module of CAM5 used in this
study, the aging process of BC is neglected by assuming the immediate mixing of BC
with other aerosol species. This assumption could lead to too much wet removal of
BC and therefore underestimation of BC concentration, absorption optical depth (Fig.
3) and direct radiative forcing. In addition, the well-mixed optical treatment in CAM5
could also cause bias in BC absorption calculation. H. Wang et al. (2014) examined
source-receptor relationships for BC and found that, with BC slow-aging treatment
included in CAM5, the source region contributions to global BC burden only perturbed
slightly compared to simulation without BC aging. Therefore, although the magnitude
of simulated BC and its optical properties could be underestimated due to
instantaneous aging of BC and uncertainty in coating structures, we expect the aging
treatment in MAM3 of CAM5 may not influence the source contributions examined in
this study. However, if the BC source-tagging technique could be implemented in
future models with explicit BC aging processes, e.g. the new four-mode version
(MAM4, Liu et al., 2016) of CAM version 5.3, with explicit optical property treatment, a
more accurate source-receptor relationships of BC in China could be presented.

Other comments:

(3) Line 70
Please describe the reason of the faster regional removal.
Response:
Revised as "BC in East Asia has a shorter lifetime than the global mean value
due to a faster regional removal (H. Wang et al., 2014), probably associated with
strong precipitation during monsoon season."

(4) Lines 168-169
Please clarify the definition of the direct radiative forcing of BC. Is this calculated from
the difference of two radiative transfer calculations with and without BC for the
clear-sky condition?
Response:
Revised as "Direct radiative forcing of BC is calculated from the difference of two
radiative transfer calculations with and without BC for the all-sky condition following
(Ghan, 2013)."

(5) Lines 182-204

Please show the difference of BC emission fluxes between the emission inventory
used in this study and other emission inventories (e.g., INTEX-B, HTAP). The values
are shown later (at Lines 534-538), but I think it is better to show them here. In
addition, please add some comments on the impact of larger values of BC emissions
in this study on the estimation of source contributions of BC. Can you add the values
of BC emissions from outside China (e.g., India, Southeast Asia, Japan, Korea) to
Figure 1b?
Response:
Thanks for the suggestion. We have added Table S1 to compare the
anthropogenic emission used in this study with emissions from previous studies. The
anthropogenic emission of BC in China for 2010–2014 is larger than those from
previous studies, partly resulting from the rapid increasing trend of BC in China during
recent years. The higher emission could lead to higher concentration and direct
radiative forcing, and source contributions of BC in China. We have added these
descriptions in methods section.
We also added BC emissions from outside China in Figure 1c. Emissions in
continental level are summarized here instead of Country level because the model
revolution is a bit course compared to country level emission. Total BC emissions
from neighboring regions including, rest of East Asia (REA, without China), South
Asia (SAS), Southeast Asia (SEA), and Russia/Belarussia/Ukraine (RBU) are shown
in Figure 1c.   The source regions outside China are consistent with regions source
regions defined in the second phase of Hemispheric Transport of Air Pollution
(HTAP2). South Asia and Southeast Asia have higher emissions, which may
contribute to concentration and direct radiative forcing of BC in China, especially
southern and western China, through long-range transport. We have also added
these in the methods section.

**Table S1.** Comparisons of annual anthropogenic BC emissions in China with
previous studies.

|  | Year | Anthropogenic emission in China (Gg/yr) |
|---|---|---|
| This study (CEDS, Hoesly et al., 2017) | 2010–2014 | 2467 |
| MIX (Li et al., 2017) | 2010 | 1765 |
| HTAP V2.2 (Janssens-Maenhout et al., 2015) | 2010 | 1741 |
| Lu et al. (2011) | 2010 | 1751 |
| Qin and Xie (2012) | 2009 | 1764 |
| Wang et al. (2012) | 2007 | 1879 |
| INTEX-B (Zhang et al., 2009) | 2006 | 1811 |

[Figure]

[Figure]

**Figure 1.** (a) Spatial distribution of annual mean total emissions (anthropogenic plus
biomass burning, units: g C m$^{-2}$ yr$^{-1}$) of black carbon (BC) averaged over 2010–2014.
The geographical BC source regions are selected as North China (NC, 109°E–east
boundary, 30°–41°N), South China (SC, 109°E–east boundary, south boundary–
30°N), Southwest China (SW, 100°–109°N, south boundary–32°N), Central-West
China (CW, 100°–109°N, 32°N–north boundary), Northeast China (NE, 109°E–east
boundary, 41°N–north boundary), Northwest China (NW, west boundary–100°E,
36°N–north boundary), and Tibetan Plateau (TP, west boundary–100°E, south
boundary–36°N) in China and regions outside of China (RW, rest of the world). (b)
Seasonal mean total emissions (units: Gg C, Gg = 10$^9$g) of BC from the seven BC
source regions in China and (c) emissions from rest of East Asia (REA without
China), South Asia (SAS), Southeast Asia (SEA), and Russia/Belarussia/Ukraine
(RBU).

(6) Lines 261-263
You can show the contributions from outside China quantitatively from the tagged
simulation results.
Response:
    We quantitatively showed the contributions from outside China in the results
section. And it is not suitable to compare surface value and burden value. Therefore
we have deleted this sentence to avoid duplicate and potential misleading.

(7) Line 281
Please describe the reason of BC underestimation by up to a factor of 20.
Response:
    We have added a sentence, "where BC concentrations appear to be
underestimated in the model (up to 20 times lower). The possible bias is discussed in
the following part", because we fully discussed this underestimation of BC in the
following part and in the discussion section, as
    "Note that the model largely underestimates BC concentrations over China,
compared to the observation, which has also been reported in many previous studies
using different models and different emission inventories (e.g., Liu et al., 2012; Fu et
al., 2012; Huang et al., 2013; H. Wang et al., 2013; Q. Wang et al., 2014; R. Wang et
al., 2014; Li et al., 2016). One possible reason is that in situ measurements are point
observations, while the model does not treat the subgrid variability of aerosols and
assumes aerosols are uniformly distributed over the grid cell. R. Wang et al. (2014)
found a reduction of negative bias (from −88% to −35%) in the modeled surface BC
concentrations when using high-resolution emissions and modeling at 0.5° X 0.7°
resolution. They find, however, that modeling over the North China Plain at an even
higher resolution of 0.1°, further reduced the surface concentration bias there from
29% to 8%. This result indicates that the siting of observational stations can result in
an artificial bias when comparing with relatively coarse model results. Further
investigation of this siting/resolution bias is warranted, including investigation if this type of bias might extend, presumably to a lesser exent, also to AAOD measurements.

Further reasons that could contribute to this bias are emission underestimation or inaccurate aerosol processes in the model. Given that the differences between modeled and observed AAOD over eastern China are relatively small (–18%), we conclude that, given current evidence, the total amount of atmospheric BC in these simulations is reasonable at least in this sub-region.

Over eastern China, the BC concentrations are dominated by local emissions in this study, with local contribution of 64–93%. The underestimation of simulated BC concentrations over eastern China is more likely due to either underestimation of local emissions, too much aerosol removal within these regions, or resolution bias between observations and model grids. Over western China, 22–76% of the BC originates from emissions outside China. Thus biases of simulated BC concentrations could also come from underestimation of emissions outside China and or too much removal of BC during long-range transport. Satellite data are a promising method to validate modeling and emissions inventories, given that they do not depend on the location of observing stations, providing more uniform spatial coverage. A comparison of modeled AAOD and satellite aerosol index (AI) provides an indication that the modeled burden in western China is underestimated, although the role of dust needs to be better characterized."

(8) Line 343
I cannot find large sources of BC in Northwest China in Figure 1a. Does the description here mean that there may be large sources of BC which are not considered in the emission inventory? Can you show the contribution of BC and dust to AAOD (in model) over this region? I think dust is dominant over this region.
Response:

Yes. The uncertainty in emission over Northwest China could be one reason, as well as too much removal in local region. In the following source attribution analysis, we found emission from outside China also significantly contribute to BC concentration in Northwest China. Therefore the underestimation in BC concentration could also come from the uncertainty in emission outside China or too much removal during long-range transport in the model.

Correct, dust is dominant over Northwest China (Fig. A). The bias in dust simulation could also lead to the difference between model and observation. That is why we notice the potential bias from dust simulation, as "It is somewhat difficult to draw a firm conclusion, however, given the likely differential role of dust, and model biases modeling dust, and possible biases in satellite derived AI values." And "A comparison of modeled AAOD and satellite aerosol index (AI) provides an indication that the modeled burden in western China is underestimated, although the role of dust needs to be better characterized."

[Figure]

AAOD (Model)

Figure A. Simulated annual mean AAOD of BC and dust.

(9) Lines 667-669

Related to the comment (2), is an internally-mixed treatment used in the calculations of AAOD? If so, AAOD should be lower (underestimated more) when more realistic

BC mixing state treatment is used in the optical calculations.

Response:

Yes, CAM5 uses internally-mixed treatment in the calculations of AAOD. We have added a caveat here, as "Note that, the model uses internally-mixed treatment in the calculations of AAOD, indicating that the AAOD could be underestimated more in the model compared observations."

[revised manuscript text omitted]